# Constrained CMIP6 projections indicate less warming and a slower increase in water availability across Asia

Yuanfang Chai [1,2], Yao Yue [1,3 ✉], Louise J. Slater [4], Jiabo Yin[1], Alistair G. L. Borthwick [5,6], Tiexi Chen[7] & Guojie Wang [7]

Climate projections are essential for decision-making but contain non-negligible uncertainty. To reduce projection uncertainty over Asia, where half the world's population resides, we develop emergent constraint relationships between simulated temperature (1970–2014) and precipitation (2015–2100) growth rates using 27 CMIP6 models under four Shared Socio-economic Pathways. Here we show that, with uncertainty successfully narrowed by 12.1–31.0%, constrained future precipitation growth rates are $0.39 \pm 0.18$ mm year$^{-1}$ (29.36 mm °C$^{-1}$, SSP126), $0.70 \pm 0.22$ mm year$^{-1}$ (20.03 mm °C$^{-1}$, SSP245), $1.10 \pm 0.33$ mm year$^{-1}$ (17.96 mm °C$^{-1}$, SSP370) and $1.42 \pm 0.35$ mm year$^{-1}$ (17.28 mm °C$^{-1}$, SSP585), indicating overestimates of 6.0–14.0% by the raw CMIP6 models. Accordingly, future temperature and total evaporation growth rates are also overestimated by 3.4–11.6% and −2.1–13.0%, respectively. The slower warming implies a lower snow cover loss rate by 10.5–40.2%. Overall, we find the projected increase in future water availability is overestimated by CMIP6 over Asia.

[1] State Key Laboratory of Water Resources and Hydropower Engineering Science, Wuhan University, Wuhan 430072, China. [2] Vrije Universiteit Amsterdam, Department of Earth Sciences, Boelelaan 1085, 1081 HV Amsterdam, the Netherlands. [3] Institute for Water-Carbon Cycles & Carbon Neutrality, Wuhan University, Wuhan 430072, China. [4] School of Geography and the Environment, University of Oxford, Oxford OX1 3QY, United Kingdom. [5] Institute for Infrastructure and Environment, School of Engineering, The University of Edinburgh, The King's Buildings, Edinburgh EH9 3JL, UK. [6] School of Engineering, Mathematics and Computing, University of Plymouth, Drake Circus, Plymouth PL4 8AA, UK. [7] School of Geographical Sciences, Nanjing University of Information Science and Technology, Nanjing 210044, China. ✉email: yueyao@whu.edu.cn

Quantification of future precipitation response to global warming is a key issue for climate change mitigation and water resources management, both of which are particularly important in Asia (Fig. 1a) where almost half of the world's population resides[1,2]. General Circulation Models have been widely applied to project precipitation in future decades under various emission scenarios[3-6], and it is generally expected that the climate in Asia will become wetter as greenhouse gas emissions rise[7-9]. However, precipitation projections vary widely across different models[10-12], due to complex spatio-temporal variability of the tropical monsoon climate, interactions between sea air and local circulations, and variability of different internal and external forcing factors (e.g. carbon emissions, solar radiation)[13-15]. Although considerable improvements have been achieved by the CMIP5 models, which reproduce precipitation projections and summer monsoon events in Asia better than the CMIP3 models[16,17], vast uncertainty persists. In the RCP8.5 scenario, mean daily precipitation in South Asia is projected to increase by 0.46–1.92 mm day$^{-1}$ between the periods of 1986–2002 and 2081–2100[18]. Such a wide range of precipitation projections generally implies considerable variations in temperature and total

evaporation in the models to maintain energy and water balances[19-21]. Uncertainty in temperature projections also affects the projected changes in snow cover. Due to their finer spatial input data and more detailed descriptions of physical and biological processes, the CMIP6 models are expected to provide more accurate estimates of past, present, and future climate changes[22,23]. Yet, uncertainty is not negligible in projections of precipitation, land surface temperature, total evaporation, and snow cover change over Asia. Therefore, there is lack of clarity regarding future rates of climate change and possible associated risks to society, including heat stress, intense rainfall or flooding, and water availability more generally.

Recently, an innovative technique called the emergent constraint has been developed to constrain uncertainty across climate model ensemble projections[24-28]. The uncertainty in model simulations can be constrained by observations to obtain more accurate projections of future climate change[29] by developing physically explainable empirical relationships between the simulated current and future climate. The emergent constraint approach relies on the development of a near-linear relationship with a sound physical basis between present and future climate

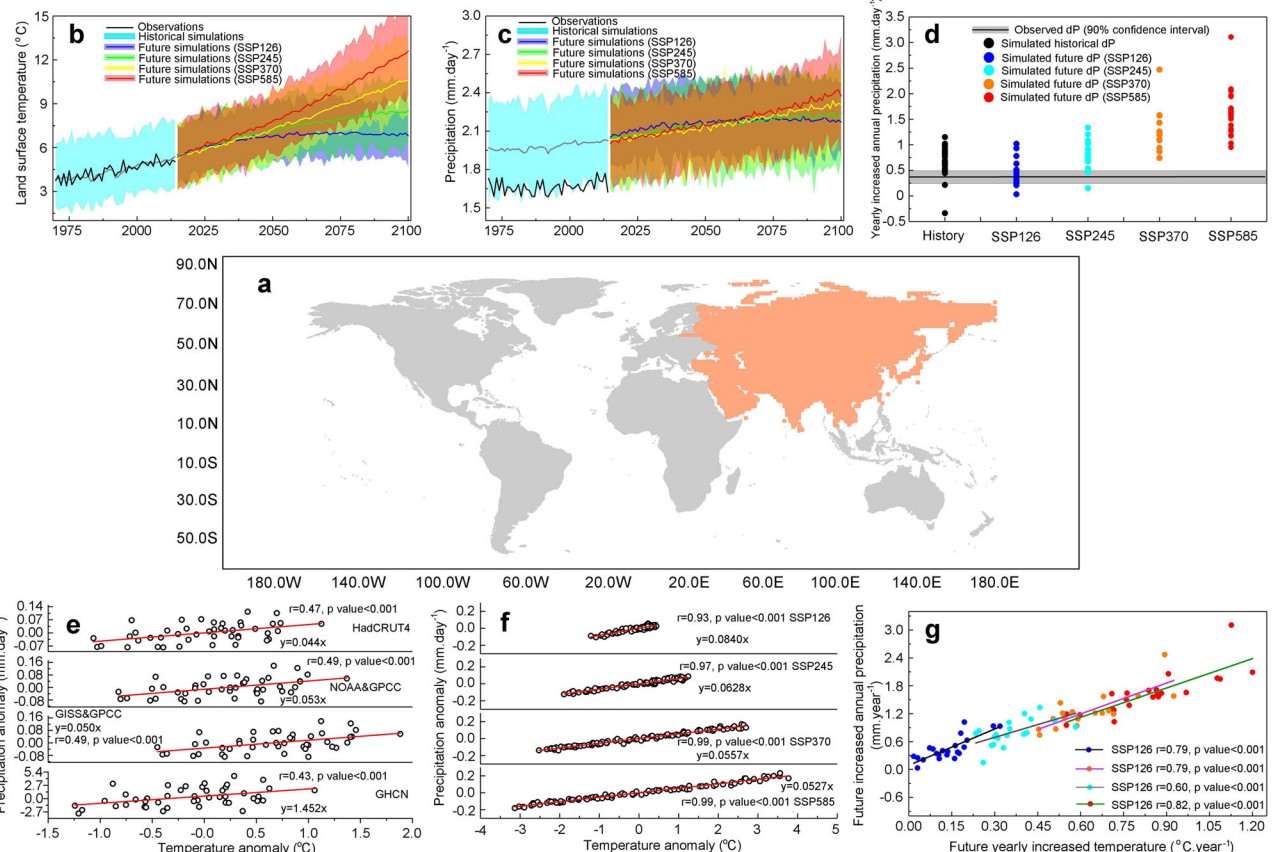

**Fig. 1 Land surface air temperature and precipitation in Asia based on observations and CMIP6 simulations. a** the orange area indicates the location of the research domain in this study. **b**, **c** are the observed and simulated changes in annual land surface air temperature (°C) and annual mean daily precipitation (mm day$^{-1}$) during the historical period of 1970–2014 and future period of 2015–2100, respectively. **d** is the range of precipitation growth rates during the historical (1970–2014) and future (2015–2100) periods under SSP126, SSP245, SSP370, and SSP585 across 27 CMIP6 models (Supplementary Table 1), calculated by fitting linear regression to the CMIP6 time series data. The horizontal gray shading represents the observed annual precipitation growth rate along with its uncertainty (90% confidence interval, 0.375 ± 0.122 mm year$^{-1}$), estimated from 1000 bootstrapped samples of the observed trends from the HadCRUT4 and GPCC data sets. Each circle represents one CMIP6 model. **e** shows the linear relations between precipitation anomalies and temperature anomalies based on the observed data sets of HadCRUT4, NOAA & GPCC, GISS & GPCC, and GHCN during 1970–2014 (Supplementary Table 2). **f** shows the linear relations between precipitation anomalies and temperature anomalies based on CMIP6 projections during 2015–2100 under SSP126, SSP245, SSP370, and SSP585. **g** shows the linear relations between the future annual precipitation growth rate and the future annual temperature growth rate based on CMIP6 projections under SSP126 ($y = 26.01x + 0.092$, $r = 0.79$, $p < 0.001$), SSP245 ($y = 18.27x + 0.145$, $r = 0.60$, $p < 0.001$), SSP370 ($y = 22.19x − 0.136$, $r = 0.79$, $p < 0.001$), and SSP585 ($y = 21.06x − 0.141$, $r = 0.82$, $p < 0.001$).

parameters. For instance, as temperature increases, enhanced longwave emission lowers radiative energy in the atmosphere[19], precipitation intensifies (due to the enhanced moisture-holding capacity of the atmosphere), and additional latent heat must be released to maintain the energy balance[30]. This process implies that a theoretically plausible constraint relationship is likely to exist between land surface temperature and precipitation, and the uncertainty of precipitation projections might be reduced based on this relationship. Here we investigate the emergent constraint relationship between the annual growth rate of historical simulated temperature during 1970–2014 and the annual growth rate of future precipitation projections during 2015–2100 in Asia using 27 CMIP6 models under SSP126, SSP245, SSP370 and SSP585 (Supplementary Table 1). Observed temperatures from the four data sets (Supplementary Table 2) are then employed to compute precipitation projections with reduced uncertainty across Asia. Constraint relationships are also explored between future changes in precipitation and temperature/evaporation, and between future changes in temperature and snow cover loss. Based on these relationships, we provide more reliable estimates of future warming and water availability in Asia, which are fundamentally important for developing policy for climate change mitigation and water resources management.

## Results and discussion

**Identification of the dominant factor.** CMIP6 models perform well at simulating the historical climate during 1970–2014 in Asia. Figure 1b shows that the difference between the CMIP6 multi-model mean value of the annual average land surface air temperature (4.43 °C, calculated as the average grid cell value of all raw model outputs across Asia, with no bias correction performed) and the observational value from the HadCRUT4 data set (4.44 °C) is only −0.01 °C. By contrast, the difference for the CMIP5 models is much higher (−0.34 °C, see Section 1 in the Supplementary Text and Supplementary Figs. 1a, 2a & b). The CMIP6 models also successfully capture the historical increasing trend of Asian annual precipitation, using the same approach (Fig. 1c). However, most of the CMIP6 and CMIP5 models overestimate historical precipitation when compared with the observations (Fig. 1c, Section 1 in the Supplementary Text and Supplementary Figs. 1b & 2c, d).

During the future period (2015–2100), the land surface air temperature in Asia presents a rising trend based on all 27 CMIP6 models under various emission scenarios (Supplementary Fig. 3a-d). By fitting linear regressions to the model time series data in Supplementary Fig. 3a-d, we estimated the mean annual temperature growth rate to be 0.0138 ± 0.0079 °C year$^{-1}$ (SSP126), 0.0358 ± 0.0092 °C year$^{-1}$ (SSP245), 0.0629 ± 0.0139 °C year$^{-1}$ (SSP370) and 0.0837 ± 0.0174 °C year$^{-1}$ (SSP585). In response to the warming climate, precipitation in Asia projected by all the 27 CMIP6 models also presents an increasing trend (Supplementary Fig. 3e-h). However, the ranges of the projected future annual precipitation growth rates across models are very wide (0.451 ± 0.258 mm year$^{-1}$ under SSP126, 0.799 ± 0.277 mm year$^{-1}$ under SSP245, 1.260 ± 0.389 mm year$^{-1}$ under SSP370 and 1.622 ± 0.444 mm year$^{-1}$ under SSP585). In addition, the annual precipitation growth rates projected by most CMIP6 models lie well above the observed range (the gray horizontal ribbon in Fig. 1d). This implies that most models have sizeable, systematic bias when simulating future precipitation[31].

In order to constrain the uncertainty in precipitation projections by CMIP6 models, the dominant factors explaining the inter-model spread must be identified to provide a physical basis for building emergent constraint relationships[32] (see Methods). We collected observed data sets of precipitation (GPCC,

20CRv2c, HadCRUT4, GHCN, CMAP, and ERA-Interim) and temperature (Delaware, HadCRUT4, GISS, and NOAA), and found positive linear relationships between the historical Asian temperature and precipitation (1970–2014) in all the data sets (Fig. 1e and Supplementary Fig. 4), indicating that precipitation changes in Asia are closely related to changes in local temperature. To examine the spatial consistency of the correlations, we randomly selected eight rectangular areas in Asia (Supplementary Fig. 5) and found significant positive relationships in each of the sub-areas, supporting the reasonability of computing the areal-mean value across all grid cells. This increase in precipitation with warming land is distinct in almost all the CMIP6 model projections (Fig. 1f and Supplementary Fig. 6), with high positive correlation coefficients ($r \geq 0.4$ and $p < 0.001$) across most (82.0–92.1%) of Asia (Supplementary Fig. 7). Thus, we can expect that future annual precipitation growth rates across models are closely and linearly related to changes in temperature, on average. By plotting the linear relation between future annual precipitation growth rates and future annual temperature growth rates (Fig. 1g, where each color circle represents the mean of one CMIP6 model), we find that the higher the temperature trend projected by a CMIP6 model, the higher the annual precipitation growth rate derived from the same model, leading to a wide spread of precipitation across models (Fig. 1d). This implies that the uncertainty in projecting the future annual precipitation growth rate across CMIP6 models is also highly dependent on the temperature simulations.

The physical mechanisms behind the linear relationship in Fig. 1g can be explained from an energy balance perspective. Surface energy balance is commonly written as Eq. (7)[33,34] (see Methods). In a closed system (i.e., no net lateral moisture influx or convergence), the absolute value of total evaporation is equal to the local precipitation over a long period. Thus, Eq. (7) is transformed to Eq. (8) (see Methods), which presents a positive change of precipitation with land surface air temperature in order to maintain the energy balance. Increased temperature also considerably enhances evaporation[35]. The resulting elevated $CO_2$ concentrations are likely to increase vegetation transpiration through a fertilization effect, as found in both observed and simulated evidence[36]. Thus, atmospheric moisture increases with evaporation and transpiration, enhancing precipitation. In warming conditions, the water-holding capacity of the atmosphere has been estimated to increase by 7% K$^{-1}$ [37] using the Clausius-Clapeyron equation, applied to evaluate the sensitivity of precipitation change to temperature variation worldwide[38,39] (see Eqs. (9, 10) in Methods). In response to the increased saturation specific humidity, precipitation is also expected to depend linearly on temperature change (increase by 1−3% K$^{-1}$), according to a thermodynamic scaling relation (Eqs. (11, 12)[37,40] in Methods). Sun et al.[6] derived similar results, with precipitation increasing by 2.5% under the 1.5 °C warming scenario in China.

However, Asia is not a 'closed system'. The Asian climate is not only affected by the thermodynamic process, but also by dynamic factors which are closely related to Asian monsoons, the El Niño Southern Oscillation, and the Arctic Oscillation[41–43], complicating the relationship between temperature and precipitation. By examining the influence of atmospheric circulation on precipitation change in Asia, we found that the dynamic factors exhibit some correlation with the long-term trend in precipitation in continental Asia ($−0.46 < r < −0.24$), but not so strong as the thermodynamic factors. Therefore, a near-linear relationship between the annual growth rates of temperature and precipitation for such a large-scale region is reasonable (see Section 2 in the Supplementary Text, and Supplementary Figs. 8–11). However, the contribution of dynamic factors will be assessed in future work.

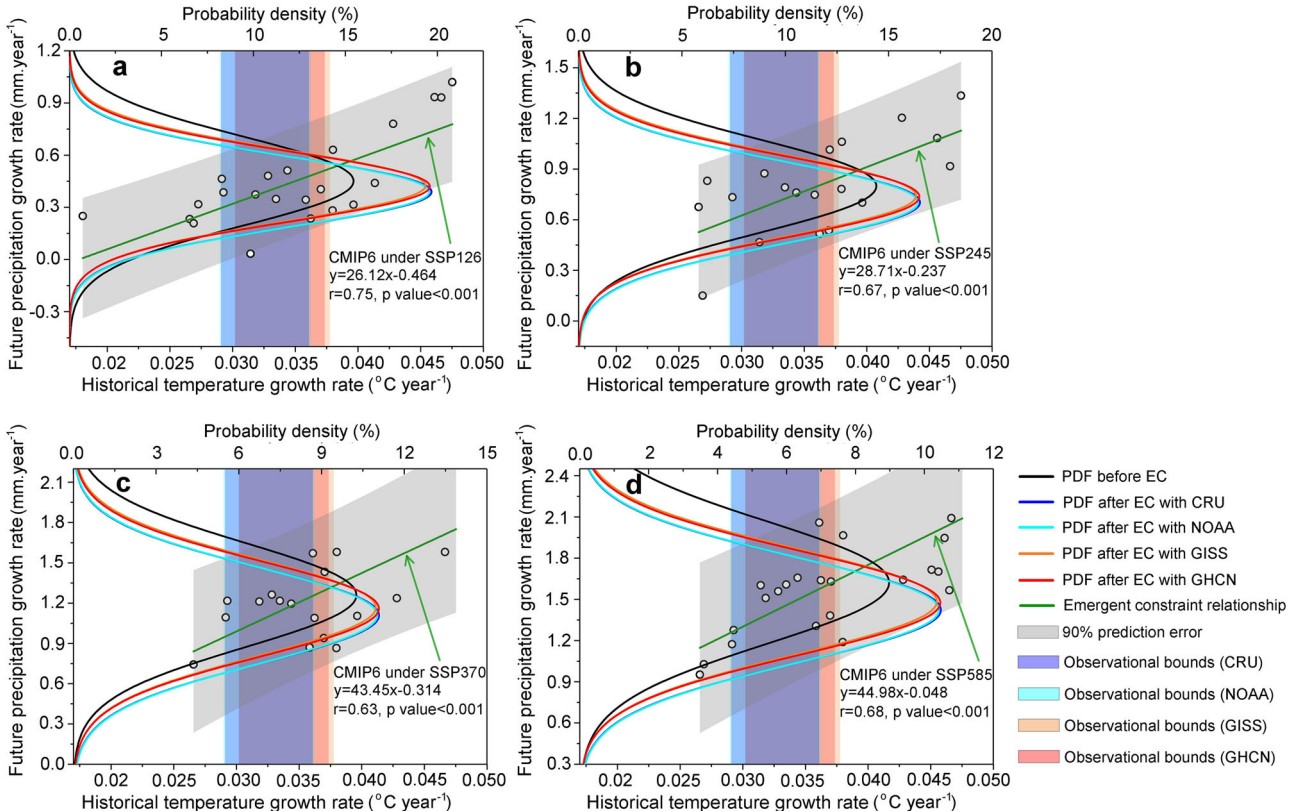

**Fig. 2 Emergent constraint (EC) on future rate of change of precipitation in Asia, based on CMIP6 projections. a–d** demonstrate the emergent constraint relationships (green line) between the simulated historical annual temperature growth rate during 1970–2015 (°C year⁻¹) and the future annual precipitation growth rate during 2015–2100 (mm year⁻¹) across 27 CMIP6 models under SSP126, SSP245, SSP370, and SSP585 emission scenarios. Each circle represents the mean values of simulated historical annual temperature growth rate and future annual precipitation growth rate from one CMIP6 model. Annual growth rates of temperature and precipitation for each model are estimated by fitting linear regression to the simulated/projected CMIP6 time series data. Observational bounds of the four observational data sets (HadCRUT4, NOAA, GISS and GHCN, see the vertical shading) are all applied (detailed values provided in Supplementary Table 8). Probability density functions of future annual precipitation growth rate are shown before (black lines) and after application of the emergent constraint to the observations (color lines).

Supported by sound physical mechanisms, a potential emergent constraint relation is expected to exist between historical temperature change and future precipitation change. Moreover, the uncertainty in future precipitation projections is expected to be narrowed by employing temperature observations in this emergent constraint relationship.

**Emergent constraint on precipitation in Asia.** Empirical relationships between the simulated historical annual temperature growth rates during 1970–2014 and the future annual precipitation growth rates during 2015–2100 across the 27 CMIP6 models for the whole of Asia (see Section 3 of the Supplementary Text and Supplementary Fig. 12) represent the emergent constraint regression equations under the four emission scenarios (Fig. 2). These four relationships are all significant ($p < 0.001$). To decrease the probability that the emergent relationship has emerged purely by chance[29], we conducted out-of-sample testing using CMIP5 ensembles. The resulting high correlations ($r = 0.72$, $p$ value < 0.001, see Supplementary Fig. 13) provide further evidence of the reliability of the emergent constraint relationships. The constrained future precipitation growth rate is then estimated by projecting the observed temperature growth rate (and its uncertainty) onto the y-axis, using the emergent constraint regression equation for each shared socioeconomic pathway (SSP). Considering the variability of observational temperature across different data sets, we obtained temperature data from four data sets (HadCRUT4, NOAA, GISS and GHCN) to improve the reliability of the estimated emergent

constraint. Probability density functions (PDFs, see Methods) fitted to a Gaussian distribution are also drawn for both the original and the constrained future annual precipitation growth rates based on all the selected CMIP6 models (Fig. 2). All the PDF curves in Fig. 2 show that the mean values of the constrained results are consistently shifted lower under the four emission scenarios using different observational data sets (i.e. reduced precipitation compared with the original CMIP6 model outputs, see Supplementary Table 8 for detailed results). The constrained results decrease from the original 0.451 mm year⁻¹ (SSP126), 0.799 mm year⁻¹ (SSP245), 1.260 mm year⁻¹ (SSP370) and 1.622 mm year⁻¹ (SSP585) to 0.388 mm year⁻¹, 0.699 mm year⁻¹, 1.102 mm year⁻¹ and 1.418 mm year⁻¹, respectively, indicating an overestimate of 6.0–14.0% by the original CMIP6 models. More importantly, the PDF curves in Fig. 2 become narrower after application of the emergent constraint. This implies that uncertainty in the projected future precipitation growth rates across the CMIP6 models (Fig. 1d) was successfully constrained, with standard deviations decreasing from 0.258 mm year⁻¹ (SSP126), 0.277 mm year⁻¹ (SSP245), 0.389 mm year⁻¹ (SSP370) and 0.444 mm year⁻¹ (SSP585) to 0.178–0.189 mm year⁻¹, 0.222–0.229 mm year⁻¹, 0.332–0.342 mm year⁻¹ and 0.350–0.368 mm year⁻¹, respectively (i.e. reduced by 12.1–31.0%, see Supplementary Table 9 for details). The Kolmogorov-Smirnov (K-S) test[44] indicates that all the shifts of PDF curves are significant at a 5% significance level (Supplementary Fig. 14 and Supplementary Table 9). Similarly, the constrained results indicate that the sensitivity of precipitation to temperature

(i.e. d$P$/d$T$) was also overestimated, decreasing from the original 32.71 mm °C$^{-1}$ (SSP126), 22.30 mm °C$^{-1}$ (SSP245), 20.03 mm °C$^{-1}$ (SSP370) and 19.37 mm °C$^{-1}$ (SSP585) to 29.36 mm °C$^{-1}$, 20.03 mm °C$^{-1}$, 17.96 mm °C$^{-1}$ and 17.28 mm °C$^{-1}$, respectively.

Strong positive changes of precipitation with temperature in accordance with thermodynamic laws during 1982–2100 are also found in other continents (e.g., North America, Europe, and Central Africa, see Supplementary Fig. 15), implying similar emergent constraint relationships might be able to provide more accurate projections of future precipitation there. By contrast, the constraint relationship is not suitable in regions where the linear coefficient between temperature and precipitation is negative (Supplementary Fig. 15). In the Amazon River basin for instance, precipitation has been found to be more sensitive to atmospheric $CO_2$ variations than to temperature changes through physiological responses of vegetation[45–47]. When $CO_2$ increases, stomatal closure has been observed[48], leading to lower total evaporation and reduced water loss during photosynthesis[49,50]. Thus, the Amazon River basin is likely to become drier under future warming scenarios. Other climate factors, such as an El Niño-like mean sea surface temperature (SST) change, may also play a key role influencing local precipitation elsewhere[51].

Other variables involved in Eq. (8) (see Methods) may also affect uncertainty in future precipitation projections. Based on outputs from the 21 CMIP6 models (Supplementary Tables 4–7), we used data on downwelling shortwave radiation, upward sensible heat flux, upwelling longwave radiation, upwelling shortwave radiation, wind speed, latent heat flux, relative humidity, soil moisture and land surface runoff to further investigate the main drivers of precipitation uncertainty. After building linear relations between the future annual precipitation growth rates and the future annual growth rates of all the above factors (Supplementary Figs. 16–24), we find that only upwelling longwave radiation is tightly related to precipitation (Supplementary Fig. 18). However, the duration of the observed upwelling longwave radiation data set is only 17 years (since 2002), and so may not be able to provide an accurate constraint. Therefore, the emergent constraint relation between the simulated historical temperature change and future precipitation change is the only choice considered in this study.

**Implications for future warming in Asia.** Although the mean temperature simulated by the models shows no significant bias against observations during the historical period, the multi-model mean annual growth rate in temperature (0.363 ± 0.0732 °C decade$^{-1}$) exhibits non-negligible bias in comparison with observations (0.326 ± 0.035 °C decade$^{-1}$); hence, the simulated temperature growth rate is overestimated by 11.35%. Probability density distributions of annual growth rates in temperature based on observations and on CMIP6 multi-model mean values also exhibit large discrepancy (see Supplementary Fig. 25). Therefore, it is legitimate to constrain the model temperature projections. As indicated by the land-surface energy balance equations (Eqs. (7), (8)) in Methods), a positive relation exists between land surface air temperature and precipitation (Fig. 1g). Thus, future temperature projections in Asia should also be adjusted to maintain consistency with the constrained precipitation projections (see rationale in Section 4 of Supplementary Text). By establishing a linear regression between the future annual precipitation growth rate and the future annual temperature growth rate under all four emission scenarios (Fig. 3a), a similar process to the emergent constraint method is undertaken. Hence, the constrained future annual precipitation growth rate can be employed to obtain more reliable estimates of future temperature change. Moreover, the

constrained future annual temperature growth rate also has lower mean and standard deviation (Fig. 3b and Supplementary Table 10) for all four emission scenarios than the original CMIP6 outputs, implying that uncertainty in projecting the future warming trend has been reduced by 13.7–29.1% (Supplementary Table 10). As shown in Fig. 3b and Supplementary Table 10, the constrained future annual temperature growth rate is expected be 0.0122–0.0131 ± 0.0056–0.0059 °C year$^{-1}$ (SSP126), 0.0338–0.0346 ± 0.0070–0.0075 °C year$^{-1}$ (SSP245), 0.0584–0.0601 ± 0.0110–0.0120 °C year$^{-1}$ (SSP370) and 0.0771–0.0791 ± 0.0139–0.0144 °C year$^{-1}$ (SSP558), indicating overestimation of 3.4–11.6% by the original CMIP6 model outputs.

To verify the reliability of the constrained results on future temperature growth rate (Fig. 3a, b), we identified another potential emergent constraint relationship between the historical observed and future projected annual temperature growth rates across the CMIP6 models. The proposed mechanism underpinning the emergent relationship is that there exists a proportionally positive response in temperature to the rising radiative forcing, i.e., past and future warming trends are both controlled by sensitivity to radiative forcing[52]. This mechanism has been widely applied to constrain equilibrium climate sensitivity (ECS), transient climate response (TCR), and ocean heat uptake[52–57]. As shown in Supplementary Fig. 27a, the emergent constraint relationships between the temperature growth rates of the historical (1970–2014) and future (2015–2100) periods are significant under the four emission scenarios. After applying the observed temperature from HadCRUT4, NOAA, GISS, and GHCN datasets in these relationships, we obtained similar results, indicating that the future annual temperature growth rate has been overestimated by 4.7–11.7% compared with the original CMIP6 projections (Supplementary Fig. 27b, c).

**Implications for future water availability in Asia.** The constrained projections of precipitation and temperature are expected to influence the projections of total evaporation and snow cover fraction, and thus, affect the estimation of future water availability in Asia. Given that total evaporation returns ~60% of land precipitation to the atmosphere[58,59], a strong constraint relation is expected between total evaporation and precipitation. By establishing a positive linear regression between future annual total evaporation and precipitation growth rates (Fig. 3c), the constrained precipitation projections were used to estimate future total evaporation changes (Supplementary Table 11), which are likely to reach 0.266–0.286 mm year$^{-1}$ (SSP126), 0.400–0.422 mm year$^{-1}$ (SSP245), 0.530–0.562 mm year$^{-1}$ (SSP370) and 0.658–0.694 mm year$^{-1}$ (SSP585), implying an overestimation of −2.1–13.0% by the original CMIP6 models. The standard deviations have been narrowed effectively across CMIP6 models by 9.2–30.1% after the application of the constraint (Fig. 3d).

The degree-day snow model equations (See Methods) show that snow cover change through melting is mainly affected by temperature and degree-day factor[60]. However, the latter does not change significantly[61]. Therefore, variation in snow cover is expected to be mainly related to temperature change[62]. By building a regression between the future annual temperature growth rate and the future annual loss of snow cover fraction (Fig. 3e), the best estimates of snow cover changes are expected to reach −0.0143 ± 0.0114% year$^{-1}$ (SSP126), −0.0377 ± 0.0137% year$^{-1}$ (SSP245), −0.0683 ± 0.0190% year$^{-1}$ (SSP370) and −0.0918 ± 0.0247% year$^{-1}$ (SSP585) when the constrained future annual temperature growth rate is applied. By comparison with the original CMIP6 projections, we find the future annual loss of snow cover fraction in Asia has been largely overestimated (by 10.5–40.2%) due to the overestimation of future temperature

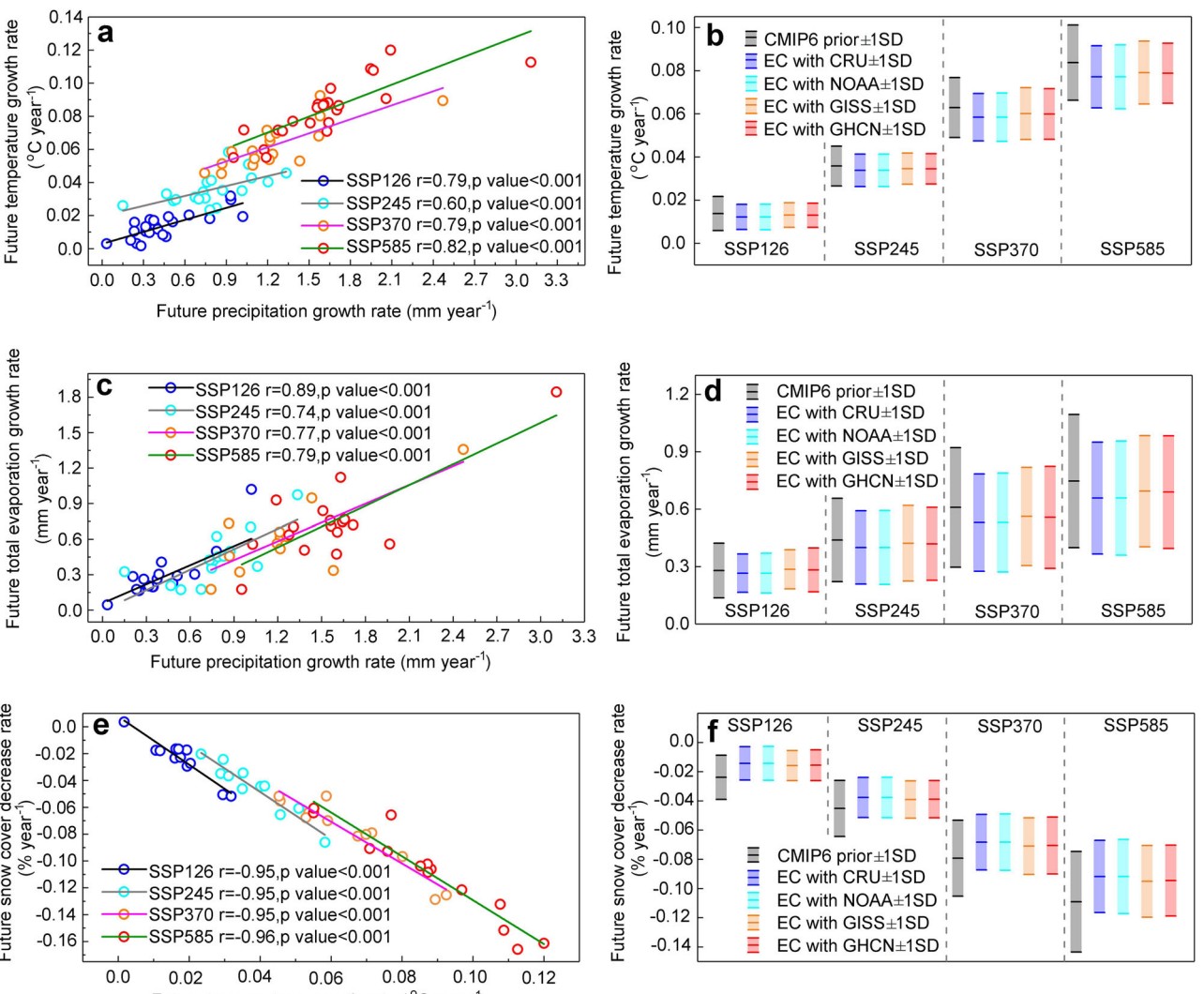

**Fig. 3 Emergent constraint (EC) on future temperature, total evaporation, and snow cover fraction changes in Asia after applying the constrained future precipitation growth rate. a** is the constraint relation between the future growth rate of precipitation (mm year$^{-1}$) and the future growth rate of temperature (°C year$^{-1}$) under SSP126, SSP245, SSP370, and SSP585 emission scenarios; (**b**) is the best estimate ± one standard deviation of the original and constrained future annual growth rate of temperature when the constrained precipitation is derived from the HadCRUT4, NOAA, GISS, and GHCN observational data sets. Similarly, (**c, e**) present constraint relationships between future annual precipitation and total evaporation growth rates (mm year$^{-1}$), and between future annual temperature and snow cover fraction (% year$^{-1}$) growth rates, respectively, with the snow-free regions of Asia excluded. **d, f** present the best estimate ± one standard deviation for future annual total evaporation and for future annual loss in snow cover fraction, respectively. Please also refer to Section 5 of Supplementary Text and Supplementary Fig. 28 for comparison between the different changing directions of the constrained results.

(Fig. 3e, f, and Supplementary Table 12). In addition, the variation range of the future annual loss in snow cover fraction across CMIP6 models has also been successfully narrowed by 23.2–33.9% (Fig. 3f). More importantly, the probability of extreme future snow cover loss in Asia is nearly zero after constraint (Supplementary Fig. 29 and Supplementary Table 13), noting that the original CMIP6 models projected 8.7%, 5.7%, 6.6% and 7.5% probability of a future annual snow cover loss rate exceeding 0.045% year$^{-1}$ (SSP126), 0.076% year$^{-1}$ (SSP245), 0.119% year$^{-1}$ (SSP370) and 0.159% year$^{-1}$ (SSP585). Here the critical snow-cover loss rates are obtained by setting the probability to be 0.5% on the PDF curves after emergent constraint.

In this study, we identified an emergent constraint relationship between simulations of historical temperature growth rates and future precipitation growth rates across 27 CMIP6 models under

the SSP126, SSP245, SSP370, and SSP585 emission scenarios. After application of the emergent constraint to the temperature observations, the uncertainty in future precipitation projections was successfully reduced. We find the original CMIP6 models considerably overestimate future annual precipitation growth rates across Asia. Importantly, the constrained future precipitation growth rate was then further applied to adjust future projections of temperature and total evaporation based on reliable constraint relationships. The adjustments indicated that future temperature growth rates are overestimated by the original CMIP6 outputs, leading to considerable overestimation of future annual losses in snow cover (i.e., less snowmelt water supply). The increases in total evaporation are equally overestimated. Considering the difficulties in deriving available freshwater data[63,64], these adjusted projections provide more realistic estimates of future water availability and suggest a slower acceleration of the water cycle than previously

estimated, with less water available in the future for use by human society, animals, and vegetation due to the overestimation of snowmelt and the growth rates of precipitation and evaporation. We should note that precipitation is affected by complicated factors. Although 'thermodynamic' factors have been widely recognized as playing the lead role in driving changes in long-term mean precipitation over large areas[65,66] (while vertical pressure velocity and CAPE have smaller correlation coefficients), dynamic factors may still be significant (Supplementary Fig. 8) under certain circumstances. Therefore, it would be worthwhile to determine the specific contributions of the dynamic factors to the long-term trend in precipitation at the continental scale in future work.

## Methods

**The emergent constraint relationship.** The emergent constraint method relies on an empirical near-linear relationship between a historical simulated variable (namely "independent variable $x$") and a future climate predicted variable (namely "dependent variable $y$") across an ensemble of models[29,32]. Observed changes in variable $x$ typically provide a more reliable trend or variation, for the measurement uncertainty in climate variable $x$ is usually small compared to the range of simulated values. Therefore, by projecting the observed variable $x$ with its uncertainty (represented by one standard deviation) onto the $y$-axis through an empirical linear relationship, it is possible to obtain more reliable future changes in the predicted variable $y$ with narrower uncertainty (see Section 6 of the Supplementary Text for comparison between emergent constraint and bias correction methods). We use the least-squares linear regression method to build the emergent constraint relationship (See Eq. (1))[25]. The prediction error of the regression ($\sigma_y$) is calculated by Eq. (2).

$$y_i = ax_i + b \tag{1}$$

where $y_i$ is the value given by $x_i$; $a$ and $b$ are the slope and intercept values, respectively;

$$\sigma_y(x) = s\sqrt{1 + \frac{1}{N} + \frac{(x - \overline{x})^2}{N \cdot \sigma_x^2}} \tag{2}$$

where $s$ is used for minimizing the least-squares error, calculated by Eq. (3); and $N$ is the number of models in the ensemble. $\sigma_x$ is the variance of $x_i$, calculated by Eq. (4); $\overline{x}$ is the mean value;

$$s^2 = \frac{1}{N-2} \sum_{n=1}^{N} (y - y_i)^2 \tag{3}$$

$$\sigma_x = \sqrt{\sum_{n=1}^{N} (x_i - \overline{x})^2 / N} \tag{4}$$

**Calculation of probability density function (PDF).** We used Eq. (5) to estimate the PDF of the original projected variable (from CMIP6) before applying the emergent constraint[25,28].

$$\text{PDF}(y/x) = \frac{1}{\sqrt{2\pi \cdot \sigma_y^2}} \exp \left\{ -\frac{(y - f(x))^2}{2\sigma_y^2} \right\} \tag{5}$$

where PDF($y/x$) is the PDF around the best-fit linear regression, representing the PDF of $y$ given $x$.

After the constraint is applied, the PDF for the constrained projected variable (PDF($F$)) is calculated by numerically integrating PDF($F/ob$) and PDF($ob$) (Eq. (6)), where PDF($F/ob$) is the probability density for the "future climate projected variable" given the "historical observable variable", and PDF($ob$) is the observation-based PDF for the "historical observable variable".

$$\text{PDF}(F) = \int_{-\infty}^{+\infty} \text{PDF}(F/ob) \cdot \text{PDF}(ob) \cdot d\,ob \tag{6}$$

**Land-surface energy balance equation.** The land-surface energy balance equation is commonly written as Eq. (7)[33,34]. In a fully closed system, Eq. (7) can be written as Eq. (8), due to equal absolute values between total evaporation and precipitation over a long period[28].

$$\lambda E + \varepsilon \sigma T_0^4 = (1 - \alpha)R_{swd} + \varepsilon R_{lwd} - G_0 - H \tag{7}$$

$$\lambda \cdot (-P) + \varepsilon \cdot \sigma \cdot T^4 = (1 - \alpha) \cdot R_{swd} + \varepsilon \cdot R_{lwd} - G_0 - H \tag{8}$$

where $\lambda$ is the latent heat of vaporization, $E$ is the actual total evaporation, $\varepsilon$ is the surface emissivity, $\sigma$ is the Stefan-Boltzmann constant, $T_0$ is the surface temperature, $\alpha$ is the albedo, $R_{swd}$ is the downward solar radiation, $R_{lwd}$ is the downward

longwave radiation, $G_0$ is the soil heat flux, $H$ is the turbulent sensible heat flux, and $P$ is the precipitation.

**Thermodynamic equations.** The Clausius-Clapeyron relation (Eqs. (9), (10))[37], which is used to examine the sensitivity of precipitation change to temperature variation worldwide including Asia[38,39], shows that increasing temperature leads to an increase in specific humidity, which can enhance precipitation, thereby resulting in a positive sensitivity of precipitation to temperature (Eqs. (11),(12))[37,40].

$$\frac{dq_s}{dT} = \frac{q_s \cdot L_v}{R_v \cdot T^2} \tag{9}$$

$$\frac{dq_s}{q_s} = \frac{L_v}{R_v \cdot T^2} dT = \alpha \cdot dT \tag{10}$$

where $q_s$ is saturation specific humidity, $T$ is temperature, $L_v$ is latent heat of condensation at temperature $T$ (assumed $2.5 \times 10^6 \, \text{J kg}^{-1}$), and $R_v$ is the gas constant for water vapor ($461.5 \, \text{J kg}^{-1} \, \text{K}^{-1}$). (Under the condition that the total pressure is much larger than the water vapor pressure, $\alpha$ is calculated to be $0.07 \, \text{K}^{-1}$, in other words, $q_s$ increases by 7% per degree of warming.)

$$\text{Pre} = M_f \cdot q_s \tag{11}$$

$$\frac{dPre}{Pre} = \frac{dM_f}{M_f} + \frac{dq_s}{q_s} = \frac{dM_f}{M_f} + 0.07 \cdot dT \tag{12}$$

where $Pre$ is precipitation, and $M_f$ is convective mass flux. Given that $M_f$ is usually assumed to be unchanged (i.e., $dM_f = 0$), it is reasonable to assume that precipitation is linearly dependent on temperature change. Equation (12) is also constrained by radiative cooling, and so the increasing rate of precipitation is expected to be weakened by 4–6% $\text{K}^{-1}$, and have a value of about 1–3% $\text{K}^{-1}$ [37,38].

**Equations from the degree-day snow model.** The main equations involved in the degree-day snow model are as follows[62,67]:

$$m = k_d(T - T_0) \, for \, T > T_0 \tag{13}$$

$$m = 0 \, for \, T \le T_0 \tag{14}$$

where $m$ is the snow melting rate (mm.day$^{-1}$), and $k_d$ is the degree day factor (mm °C$^{-1}$.day$^{-1}$). Being mainly affected by changes in snow properties, the value of $k_d$ usually does not change significantly in a season. $T$ is the surface temperature, and $T_0$ is the base temperature (usually, 0 °C).

## Data availability

The shapefile of the Asian boundary is from https://www.naturalearthdata.com/downloads/110m-cultural-vectors/110m-admin-0-countries/.

Simulated data on precipitation and temperature during 1970–2100 and on total evaporation, downwelling longwave radiation, downward sensible heat flux, downwelling shortwave radiation, surface upward sensible heat flux, surface upwelling longwave radiation, surface upwelling shortwave radiation, and wind during 2015–2100 from CMIP6 models were collected from https://esgf-node.llnl.gov/projects/cmip6/. Simulated data on precipitation and temperature during 1970–2100 from CMIP5 models were collected from https://esgf-node.llnl.gov/search/cmip5/.

Observed precipitation and temperature data were obtained from the data sets of HadCRUT4 (http://www.cru.uea.ac.uk/), GHCN (https://www.ncdc.noaa.gov/ghcn-monthly), NOAA (https://www.esrl.noaa.gov/psd/data/gridded/data.noaaglobaltemp.html), GISS (https://www.esrl.noaa.gov/psd/data/gridded/data.gistemp.html) and GPCC (https://climatedataguide.ucar.edu/climate-data/gpcc-global-precipitation-climatology-centre). We regridded all CMIP6 outputs and observational data sets to a common 0.25° × 0.25° latitude-longitude spatial resolution to calculate the CMIP6 multi-model mean values.

## Code availability

The code for this study is available by request from the corresponding author.

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

## Acknowledgements

Y.Y. acknowledges support from the National Natural Science Foundation of China (Grant No. 52079094), the National Key R&D Program of China (Grant No. 2016YFA0600901), and the Youth Project of the National Natural Science Foundation of China (Grant No. 41601275). Y.C acknowledges support from the China Scholarship Council. L.J.S. acknowledges support from UK Research and Innovation (Grant No. MR/V022008/1). J.Y. acknowledges support from the Youth Project of the National Natural Science Foundation of China (Grant No. 52009091). A.G.L.B. acknowledges support from UK NERC Global Challenges Research Fund (Grant No. NE/S009000/1). T.C. acknowledges support from the National Natural Science Foundation of China (Grant Nos. 42161144003 and 42130506). G.W. acknowledges support from the National Natural Science Foundation of China (Grant No. 41875094).

## Author contributions

Y.Y. and Y.C designed the research and performed the data analysis. Y.C. and Y.Y. led the writing. L.J.S, J.Y., A.G.L.B., T.C and G.W. revised the manuscript and provided valuable comments.

## Competing interests

The authors declare no competing interests.
