## [Peer Review File · Nature Communications]

Constrained CMIP6 projections indicate less warming and a slower increase in water availability across AsiaREVIEWER COMMENTS

Reviewer #1 (Remarks to the Author):

General comments:

This study indicated slower warming and decreased water availability over Asia using CMIP6 results modified by the emergent constraint. The results of this study can be fruitful to reduce uncertainty in climate change projections over Asia. However, I can't entirely agree with the linear relationship between precipitation and temperature over Asia used in the emergent constraint method. In addition, the authors did not define the analysis region indicating Asia, where the relationship between precipitation and temperature varies depending on the area and season. Therefore, I do not recommend the publication of this manuscript until my concerns are sufficiently resolved or addressed. The details are shown below:

Major concerns:

1. The emergent constraint basically requires the linear relationship between two variables (i.e., temperature and precipitation). However, I am unsure whether there is a positive correlation between interannual variabilities of temperature and precipitation from observation. For example, more precipitation can decrease the temperature by increasing latent heat flux or increasing specific heat in land, indicating negative feedback. The authors should clarify the physical relationship between temperature and precipitation over Asia in more detail.
2. In Figure 1e, the authors showed a positive correlation between temperature anomaly and precipitation anomaly. I just wonder all models of CMIP6 showed the same relationship? It is possible that several models with strong relationship dominantly determines the overall relationship between temperature and precipitation over Asia. Also, the authors should clarify that such models with dominant relationships have better performances to simulate the Asian climate.
3. The authors never mentioned how to calculate the area-mean values for Asia. This is significantly essential because it is possible that some regions dominantly determine the linear relationship between precipitation and temperature. In Supplementary Figures 2 and 3, warming tendency over Asia is more prominent in higher latitudes, while increasing precipitation is more robust in lower latitudes. Then, how can the authors explain the linear relationship between two variables for different regions of Asia?
5. The authors elucidated the change in precipitation using the energy budget equation. However, this equation only explains the local water cycle. In fact, the Asian climate is significantly affected by a number of climate factors such as monsoon, ENSO, and AO, which make the relationship between temperature and precipitation complicated.
6. To reduce the uncertainties of climate change projection, bias correction methods have been widely used. Then what are the advantages of the emergent constraint? Does the emergent constraint lead to better performance than bias correction?

Minor concerns:

1. What is the latitude and longitude information for the Asian domain applied in this study? There is no information about this in the main text.
2. I think Supplementary Tables 1 and 2 are the same. Then what about removing Table 2?
3. Line 64-65: The authors mentioned CMIP6 models reduced temperature bias compared to the CMIP5 models. However, CMIP6 models simulated precipitation better than CMIP5 models (See Figure S1b vs. Figure 1b). Why?
4. Line 132: What is EC? Does it mean the emergent constraint? Then use the full name.
5. In Supplementary Figures. 2-4, some regions have a negative relationship between temperature and precipitation (e.g., Pakistan and Indonesia). The authors should clarify whether such regions were included in the analysis or not.
6. Line 225-227: In Figure 3b,d,f, inter-model uncertainties decreased effectively after the emergent constraint. However, the minimum values of inter-model uncertainties of temperature and evaporation did not significantly change after the emergent constraint (Figure 3b, d). In case of the snow cover, the opposite occurred. Maybe only certain CMIP6 models are sensitive to the emergent constraint, resulting in these results. Therefore, the authors should investigate the sensitivity of each model to the emergent constraint.
7. Line 228-239: There are some snow-free regions in Asia (e.g., Southeast Asia near the

equator), so the analysis region should narrow down when you calculate the relationship between snow cover fraction and temperature.

Reviewer #2 (Remarks to the Author):

Review for the paper entitled "Constrained CMIP6 projections indicate slower warming rates and reduced water availability across Asia"

The paper is well written and clear on the results. The methodology is explained in detail with all information needed. The findings are significant and seem supported by the figures, but I have a main major concern about their interpretation. Most of the emergent constraint signal is claimed to be related to precipitation feedback on temperatures. However by reading the picture my first conclusion would be that models warming the most also see the largest increase in precipitation simply because of thermodynamics laws (higher temperature = more evaporative world). For me it's not really clear that there is a feedback here. Maybe one thing to do would be to express the precipitation change per °C (instead of per year).

That's my only major concern about the paper but I think it really needs to be clarified (either by clearly showing how it's a feedback and not just thermodynamics, or by rephrasing the text and discussion around thermodynamics considerations). It would probably not change the main results (the relation between model warming the most and having more precipitation would still be the same), but it must be clear why it is so.

Some other suggestions:

Line 65: "Asia" domain should be defined here maybe?

L70: "Asian precipitation": do you consider yearly precip or only during the summer monsoon?

L137-143: Maybe a K-S test (or other) could be used to see how significant is the shift between pdf?

L164-170: It's not surprising to see a relationship between longwave and temperature, because longwave is basically temperature. Maybe you could look at the latent heat fluxes here too, which is more related to evaporation/precipitation.

L191-195: Instead of constraining the future temperatures based on a constrained precipitation why not simply simply constrain future T based on historical T?

Fig.1c: It would be nice to add some robustness/uncertainty on OBS trends (by bootstrapping the sampling for example and recomputing the trend...)

Reviewer #3 (Remarks to the Author):

With a similar approach done by Chai et al. (2021) for the Amazon dieback, this manuscript investigated historical temperature and precipitation trends in Asia by CMIP6 models against the observations, and used the results to adjust future projections. As CMIP6 models overestimate historical precipitation changes during 1970–2014, future precipitation projections are downgraded. However, the reviewer has some concerns. First, the emergent constraint seeks physical mechanism of a possible relationship between the two variables, but this approach will not work well in this subject, because regional precipitation is not governed by temperatures only, but also much influenced by circulation changes and associated moisture convergence. Second, the authors adjust future precipitation changes in the models in which historical temperature growth rate is large, although the ensemble mean historical temperature trend fits the observation very well. Third, after reducing the future precipitation trends, authors decrease the future temperature

trends, although models had no significant bias in their historical period. The reviewer is not confident with the authors hypotheses.

Other comments

(1) Methods for calculation are missing. How are different horizontal resolutions of observations and models treated? Which is the area of Asia?

(2) Line 155-159: Over the Amazon, is physiological response the main reason of precipitation decrease in future, rather than an El Nino-like mean SST changes?

(3) What is the difference between Supplementary Table 1 and 2? Captions and contents are same.

Responses to Reviewer #1

General comments:

This study indicated slower warming and decreased water availability over Asia using CMIP6 results modified by the emergent constraint. The results of this study can be fruitful to reduce uncertainty in climate change projections over Asia. However, I can't entirely agree with the linear relationship between precipitation and temperature over Asia used in the emergent constrain method. In addition, the authors did not define the analysis region indicating Asia, where the relationship between precipitation and temperature varies depending on the area and season. Therefore, I do not recommend the publication of this manuscript until my concerns are sufficiently resolved or addressed. The details are shown below:

Response: We thank the referee very much for their comments. We have addressed all the referee's suggestions, which were very valuable for improving the manuscript. In particular, we have provided further evidence, with emphasis on both the spatial differences and the inter-model differences, supporting the use of a linear relationship. Explanations of the physical mechanisms are also strengthened. The coverage of Asia has also been clearly defined, and with a new figure panel depicting the Asian-domain (in Figure 1).

Major concerns:

Comment 1. The emergent constraint basically requires the linear relationship between two variables (i.e., temperature and precipitation). However, I am unsure whether there is a positive correlation between interannual variabilities of temperature and precipitation from observation. For example, more precipitation can decrease the temperature by increasing latent heat flux or increasing specific heat in land, indicating negative feedback. The authors should clarify the physical relationship between temperature and precipitation over Asia in more detail.

Response: The referee raised a fundamental question whether a positive correlation exists between temperature and precipitation. To respond, we have provided additional evidence supporting this linear relationship in the revised manuscript, and further clarified the physical mechanisms over Asia in detail.

First, we collected additional observed datasets of precipitation (GPCC, 20CRv2c, HadCRUT4, GHCN, CMAP and ERA-Interim) and temperature (Delaware, HadCRUT4, GISS, NOAA), and found positive linear relationships between the historical temperature and precipitation (1970–2014) over Asia in all the datasets (see Supplementary Figure 4 in the revised SI and Lines 87-91 in the revised main text). To examine the correlations in detail, we randomly selected eight square areas in Asia (see Supplementary Figure 5 in the revised SI and Lines 91-94 in the revised main text), and found significant positive relationships in each of the sub-areas.

Second, we extracted future projections (2015–2100) of precipitation and temperature in Asia from each of the 27 CMIP6 models (see Supplementary Figure 6 in the revised SI and Lines 94-96 in the revised main text), and found consistent positive linear correlations in almost all the models under the four SSPs (the only exception is the NorESM2-LM model which exhibits poor performance in reproducing historical precipitation under SSP126 – the correlation coefficient between simulated and observed precipitation during 1970–2015 is only of 0.09, and the P value is 0.56), with high positive correlation coefficients ($R \geq 0.4$ and

corresponding P value < 0.001) across most (82.0–92.1%) of Asia (see Supplementary Figure 7 in the revised SI and Lines 96-97 in the revised main text).

Finally, we added in-depth explanations for the physical mechanisms behind the constraint relationship as follows:

As temperatures rise, evaporation from soil and open waters increases¹, and elevated CO₂ concentrations are expected to increase vegetation transpiration through a fertilization effect, found in both observed and simulated evidence². Thereby, atmospheric moisture increases with evaporation and transpiration, leading to more precipitation.

Further, under warming conditions, the water-holding capacity of the atmosphere is estimated to increase by 7% K⁻¹³, using the Clausius-Clapeyron equation (Eq. R1 and the converted form, Eq. R2), which has been widely used to discuss the sensitivity of precipitation change to temperature variation all over the world, including Asia⁴⁻⁵:

$$\frac{dq_s}{dT} = \frac{q_s \cdot L_v}{R_v \cdot T^2} \quad (R1)$$

$$\frac{dq_s}{q_s} = \frac{L_v}{R_v \cdot T^2} dT = \alpha \cdot dT \quad (R2)$$

where q_s and T are saturation specific humidity and temperature, respectively. L_v is the latent heat of condensation at temperature T (assumed to be 2.5×10^6 J kg⁻¹), and R_v is the gas constant for water vapor (461.5 J kg⁻¹ K⁻¹). (Under the condition that the total pressure is much larger than the water vapor pressure, α is calculated to be 0.07 K⁻¹, i.e., q_s increases by 7 % per degree of warming.)

In response to the increased saturation specific humidity, precipitation is thus expected to increase, according to a thermodynamic scaling equation (Eq. R3⁶, and the converted form, Eq. R4³):

$$Pre = M_f \cdot q_s \quad (R3)$$

$$\frac{dPre}{Pre} = \frac{dM_f}{M_f} + \frac{dq_s}{q_s} = \frac{dM_f}{M_f} + 0.07 \cdot dT \quad (R4)$$

where Pre is precipitation, and M_f is the convective mass flux. Since M_f is usually assumed to be unchanged (i.e., $dM_f=0$), precipitation is linearly dependent on temperature change. Considering that Eq. R4 is also constrained by radiative cooling, the increasing rate of precipitation is expected to be weakened by 4–6% K⁻¹, to about 1–3% K⁻¹. In agreement with this estimation, Sun et al. (2017)⁷ showed that annual precipitation in China is projected to be approximately 2.5% higher under the 1.5 °C warming scenario compared with the present-day baseline (1980–2009).

These detailed explanations on the physical mechanisms behind the linear relationship between precipitation and temperature have been added in the revised version (see Lines 110-121 in the revised main text and Lines 357-374 in Method).

Comment 2. In Figure 1e, the authors showed a positive correlation between temperature anomaly and precipitation anomaly. I just wonder all models of CMIP6 showed the same relationship? It is possible that several models with strong relationship dominantly determines the overall relationship between temperature and precipitation over Asia. Also, the authors should clarify that such models with dominant relationships have better performances to simulate the Asian climate.

Response: To address this query, we extracted future projections (2015–2100) of precipitation and temperature in Asia from each of the 27 CMIP6 models (see Supplementary Figure 6 in the revised SI), and found that almost all the models exhibit positive linear correlations between temperature and precipitation under all the SSPs (the only exception is the NorESM2-LM model which exhibits poor performance in reproducing historical precipitation under SSP126 – the correlation coefficient between simulated and observed precipitation during 1970-2015 is only of 0.09, and the P value is 0.56). The correlation coefficients are generally higher than 0.6 (P value<0.001), especially under SSP585. Therefore, our analysis does not support the assumption that only several models with strong relationships dominantly determine the overall relationship between temperature and precipitation over Asia. We have added text explaining that this is a consistent relationship across all CMIP6 models (see Lines 94-96 in the revised main text).

Comment 3. The authors never mentioned how to calculate the area-mean values for Asia. This is significantly essential because it is possible that some regions dominantly determine the linear relationship between precipitation and temperature. In Supplementary Figures 2 and 3, warming tendency over Asia is more prominent in higher latitudes, while increasing precipitation is more robust in lower latitudes. Then, how can the authors explain the linear relationship between two variables for different regions of Asia?

Response: We added the method for calculating the area-mean values for Asia as follows (see Lines 65-66 and 69-70 in the revised main text): The area-mean precipitation/temperature for the whole of Asia is calculated by averaging the values of all the grid cells, which is in agreement with other references using the emergent constraint method at global and continental scales⁸⁻¹¹ and other studies examining the sensitivity of precipitation, runoff, and evaporation¹²⁻¹⁵.

To respond to the referee's concern that some regions may dominantly determine the linear relationship between precipitation and temperature, we randomly selected eight square areas in Asia located at different latitudes (see Supplementary Figure 5 in the revised SI and Lines 91-94 in the revised main text), and found significant positive relationships during 1970–2014 in each of the sub-areas based on various observed datasets of precipitation (GPCC, 20CRv2c, HadCRUT4, GHCN, CMAP and ERA-Interim) and temperature (Delaware, HadCRUT4, GISS, NOAA) ($R>0.54$ P value<0.001). Furthermore, we examined the spatial distribution of the correlation coefficient during the future period (2015–2100) based on the CMIP6 projections, and found high positive correlation coefficients ($R\geq 0.4$ and corresponding P value<0.001) between temperature and precipitation in 82.0–92.1% of Asia, covering various latitudes (see Supplementary Figure 7 in the revised SI and Lines 96-97 in the revised main text). Therefore, the linear relationship is overall robust over Asia.

The referee is right that warming tendency over Asia is more prominent in higher latitudes with higher increasing rates of precipitation in Figs. S2 and S3. However, we should notice that, in a positive relationship, high values of one variable map high values of the other variable, and low values map low values. Consistently, in the lower latitudes of Asia where the warming tendency is relatively weak, the increasing rate of precipitation is also small. Situations in higher and lower latitudes of Asia are just the two sides of one coin.

The consistent strong positive relationships between precipitation and temperature across Asia further support the notion that it is reasonable to average the

values of all the grid cells to calculate the mean values. We have added explanations to clarify the reasonability of our method (see Line 94 in the revised main text).

Comment 4. The authors elucidated the change in precipitation using the energy budget equation. However, this equation only explains the local water cycle. In fact, the Asian climate is significantly affected by a number of climate factors such as monsoon, ENSO, and AO, which make the relationship between temperature and precipitation complicated.

Response: The referee raised a key question how other climate factors, such as monsoons, ENSO and AO, etc. affect the relationship between temperature and precipitation by altering the likelihood and intensity of extreme climate events¹⁶⁻¹⁹. To address the referee's concern, we examined the precipitation-temperature relations using a moving average method with window lengths of 5–10 years in the revised manuscript, which significantly reduces the influence of large-scale climate variability and better reflects the long-term trend²⁰⁻²¹. The results show that strong positive relations still exist between precipitation and surface air temperature after smoothing out extreme fluctuations (see Supplementary Fig. 8a in the revised SI), proving the reliability of the previously identified relationships. Furthermore, the sensitivity of precipitation to temperature change estimated by the new relationship remains (it is only slightly increased by 8.9–10.1%, see Supplementary Fig. 8b in the revised SI), implying that the effects of monsoons, ENSO and AO are not significant in this study. We added explanations in the revised manuscript (see Lines 126-137 in the revised main text).

Comment 5. To reduce the uncertainties of climate change projection, bias correction methods have been widely used. Then what are the advantages of the emergent constraint? Does the emergent constraint lead to better performance than bias correction?

Response: This is a very interesting question which, to our knowledge, has not yet been discussed by any papers. Essentially, both approaches consider the difference between projections and observations; they both assume that the difference between projections and observations over a historical period is likely to be the same in the future. However, we think the major advantage of the emergent constraint method is that it is more physically-based than bias correction because it assumes that the physics (i.e., the relationship between different variables) remain the same in the historical and future periods, while most bias correction methods simply apply a “shift” to the data. In implementing the emergent constraint method, we cannot claim that we identify an emergent constraint relationship without the support of physical mechanisms, even though we find a tight relationship between the simulated historical changes of one climate variable and the projected future changes of another²². In contrast, bias correction methods have been developed to adjust or downscale simulated climate variables²³⁻²⁴ based on correction factors that are obtained by simply exploring the statistical differences between simulations and observations.

Without sufficient emphasis on the physical mechanisms, bias correction methods sometimes destroy the physical consistency among different climate variables, leading to failures in correcting the simulated results. For instance, temperature may become sub-zero after bias correction, but rainfall is not automatically converted into snowfall²⁵. Thus, the emergent constraint method is

more reliable than the bias correction methods. We have added comparisons between the two in the revised Method (see Lines 282-283) and Section 4 of the Supplementary Text in the revised SI.

Minor concerns:

Comment 1. What is the latitude and longitude information for the Asian domain applied in this study? There is no information about this in the main text.

Response: Thank you for your comment. We have added the coverage of Asia applied in this study in Fig. 1a in the revised main text. Supplementary Figures. 2 and 7 in the updated SI are also replaced with the Asian domain versions.

Comment 2. I think Supplementary Tables 1 and 2 are the same. Then what about removing Table 2?

Response: Thanks for pointing this out. We have removed Supplementary Table 2 in the revised SI.

Comment 3. Line 64-65: The authors mentioned CMIP6 models reduced temperature bias compared to the CMIP5 models. However, CMIP6 models simulated precipitation better than CMIP5 models (See Figure S1b vs. Figure 1b). Why?

Response: We added detailed comparisons between the performances of CMIP6 and CMIP5 models in reproducing historical temperature and precipitation, and explained the reasons (see Lines 68 and 71-72 in the revised main text, Section 1 of the Supplementary Text and Supplementary Figure 2 in the revised SI):

The land surface air temperature during 1970–2014 is very well simulated by the CMIP6 models relative to observations with an error of $-0.01 \text{ }^{\circ}\text{C}$ ($4.43 \text{ }^{\circ}\text{C}$ calculated from the average grid cell value of all raw CMIP6 model outputs across Asia, with no bias correction performed vs. $4.44 \text{ }^{\circ}\text{C}$ for the observational HadCRUT4 data set). In comparison, the discrepancy between the CMIP5 outputs and the observations is much higher ($-0.34 \text{ }^{\circ}\text{C}$). Here we further estimated the mean absolute error, and found that the CMIP6 models still exhibit better performance with a mean absolute error of $1.52 \text{ }^{\circ}\text{C}$ than the CMIP5 models (mean absolute error of $1.77 \text{ }^{\circ}\text{C}$). Spatial distributions of the difference between the simulated and the observed historical temperature for CMIP6 (Supplementary Figure 2a) and CMIP5 (Supplementary Figure 2b) during the mutual historical period of CMIP5 and CMIP6 (i.e. 1986–2005) also show that the performance of the CMIP6 models is generally better than the CMIP5 outputs, especially in the Himalaya Mountains and North India.

In reproducing historical precipitation, although the mean bias of the CMIP5 models is smaller than that of the CMIP6 models (0.23 mm day^{-1} vs. 0.32 mm day^{-1}), the performance of CMIP6 models is actually better than the CMIP5 models when comparing the mean absolute bias ($0.432 \text{ mm day}^{-1}$ vs. $0.441 \text{ mm day}^{-1}$). Spatial distributions of the CMIP6-based difference between the simulated historical temperature and the observed one (Supplementary Figure 2c) and the CMIP5-based difference (Supplementary Figure 2d) during the mutual historical period of CMIP5 and CMIP6 (i.e. 1986–2005) also show that the performance of the CMIP6 models is generally better than the CMIP5 outputs, especially in the central and northern areas of India, the eastern area of Southeast Asia, and Southeast China.

Overall, the CMIP6 models have better performances in reproducing historical temperature and precipitation in Asia, because the latest generation of ESMs (CMIP6) has increased both the vertical and horizontal spatial resolutions in the models, and includes more comprehensive numerical experimental designs and more detailed processes descriptions.

Comment 4. Line 132: What is EC? Does it mean the emergent constraint? Then use the full name.

Response: Yes, EC stands for emergent constraint. We used the full name throughout the revised manuscript (Lines 145, 517, and 529). Thank you for your suggestion.

Comment 5. In Supplementary Figures. 2-4, some regions have a negative relationship between temperature and precipitation (e.g., Pakistan and Indonesia). The authors should clarify whether such regions were included in the analysis or not.

Response: As the referee pointed out, our study area included the regions presenting negative relationship between temperature and precipitation (Supplementary Figure 7), which may better reflect the complexity of the atmosphere-land-ocean interactions in Asia. However, these regions only occupy 7.9–18% (depending on the SSPs) of the total area of Asia, and we find that such a small percentage area does not affect the area-averaged emergent constraint relationship. In the revised manuscript, we discussed the inclusion of the regions with a negative relationship between temperature and precipitation, and we examined the emergent constraint relationship without these regions. We found that the positive emergent constraint relationships remained under almost all four SSPs with changes in the regression slopes of only 2.8–4.0% (Supplementary Figure 9), proving that the influence of such regions on the area-averaged emergent constraint relationship is relatively small (see Lines 141-142 in the revised main text and Section 2 of the Supplementary Text in the revised SI).

Comment 6. Line 225-227: In Figure 3b, d, f, inter-model uncertainties decreased effectively after the emergent constraint. However, the minimum values of inter-model uncertainties of temperature and evaporation did not significantly change after the emergent constraint (Figure 3b, d). In case of the snow cover, the opposite occurred. Maybe only certain CMIP6 models are sensitive to the emergent constraint, resulting in these results. Therefore, the authors should investigate the sensitivity of each model to the emergent constraint.

Response: In Figure 3b, d, f, the upper and lower lines of the colour histograms do not stand for the minimum/maximum values, but the mean \pm one standard deviation. With constrained standard deviations (i.e. $0 < SD' < SD$), when the mean value shifts downwards (i.e. $mean' < mean$, like the situations of Figure 3b and d), the lower line (indicating Mean Value – one Standard Deviation) descends less than the upper line (representing Mean Value + one Standard Deviation), because $|(mean' - SD') - (mean - SD)| < |(mean' + SD') - (mean + SD)|$. Similarly, when the mean value shifts upwards (i.e. $mean' > mean$, like the situation of Figure 3f), the upper line ascends less than the lower.

The referee also suggested that the selection of models might affect the variation range of different climate variables, which is very constructive. However, it is impossible to examine the sensitivity of individual models, because the emergent constraint relationship must be set up based on a set of models (taking Figure 2 for

example, every circle stands for one model, and a regression line is derived based on all the circles.)

Therefore, we try to analyse the effect of model selection by selecting the same models involved in developing the relationships between future total evaporation/snow cover fraction growth rates and future precipitation growth rates (listed in Supplementary Table 3), and derived new constraint relationships (see Supplementary Figure 24). The similarity between the results shown in Supplementary Figure 24b,d and Figure 3d,f suggest that the selection of models does not change the relative variation range of (Mean Value + one Standard Deviation) to that of (Mean Value – one Standard Deviation). We provided explanations in Lines 539-541 in the revised main text, and Section 3 of the Supplementary Text in the revised SI).

Comment 7. Line 228-239: There are some snow-free regions in Asia (e.g., Southeast Asia near the equator), so the analysis region should narrow down when you calculate the relationship between snow cover fraction and temperature.

Response: In this study, we focused purely on the regions with snow cover when constraining the future decreasing rate of snow cover fraction in Asia (see Fig. 3e and f in the revised main text). In the revised manuscript, we clarify that we have already excluded the snow-free regions (see Lines 537-538 in the revised main text).

References

1. Douville, H., Ribes, A., Decharme, B., Alkama, R. & Sheffield, J. Anthropogenic influence on multidecadal changes in reconstructed global evapotranspiration. *Nat. Clim. Chang.* **3**, 59-62 (2013).
2. Lobell, D.B., & Field, C.B. Estimation of the carbon dioxide (CO₂) fertilization effect using growth rate anomalies of co₂ and crop yields since 1961. *Global Change. Biol.* **14**, 39-45 (2007).
3. Kjellsson, J. Weakening of the global atmospheric circulation with global warming. *Clim. Dynam.* **45**, 975-988 (2015).
4. Li, Z., Tao, H., Hartmann, H., Su, B., & Jiang, T. Variation of projected atmospheric water vapor in central Asia using multi-models from CMIP6. *Atmosphere.* **11**, 909 (2020).
5. Wang, R., Xian, T., Wang, M., Chen, F., Yang, Y., Zhang, X., Li, R., Zhong, L., Zhao, C., Fu, Y. Relationship between Extreme Precipitation and Temperature in Two Different Regions: The Tibetan Plateau and Middle-East China. *J. Meteorol. Res.* **33**, 870–884 (2019).
6. Held, I. M., & Soden, B. J. Robust responses of the hydrological cycle to global warming. *J. Climate.* **19**, 5686-5699 (2006).
7. Sun, H., Wang, A., Zhai, J., Huang, J., Wang, Y., & Wen, S., et al. Impacts of global warming of 1.5°C and 2.0°C on precipitation patterns in China by regional climate model (cosmo-clm). *Atmos Res.* **203**, 83-94 (2017).
8. Cox, P., Huntingford, C. & Williamson, M. Emergent constraint on equilibrium climate sensitivity from global temperature variability. *Nature.* **553**, 319-322 (2018).

9. Cox, P., Pearson, D., Booth, B. et al. Sensitivity of tropical carbon to climate change constrained by carbon dioxide variability. *Nature*. **494**, 341–344 (2013).
10. Terhaar, J., Kwiatkowski, L. & Bopp, L. Emergent constraint on Arctic Ocean Acidification in the twenty-first century. *Nature*. **582**, 379-383 (2020).
11. Nijssen, F., Cox, P. M., & Williamson, M. S. (2020). An emergent constraint on Transient Climate Response from simulated historical warming in CMIP6 models. *Earth Syst Dynam.* 1-14 (2020)
12. Polson, D., Hegerl, G. C., & Solomon, S. Precipitation sensitivity to warming estimated from long island records. *Environ Res Lett.* **11**, 074024 (2016).
13. Roxy, M. Sensitivity of precipitation to sea surface temperature over the tropical summer monsoon region-and its quantification. Climate dynamics: Observational, theoretical and computational research on the climate system. *Clim. Dynam.* **43**, 1159-1169 (2014).
14. Zhang, X. Runoff sensitivity to global mean temperature change in the cmip5 models. *Geophysical Research Letters*. **41(15)**, 5492-5498 (2014).
15. Goyal, R. K. Sensitivity of evapotranspiration to global warming: a case study of arid zone of rajasthan (india). *Agricultural Water Management*. **69(1)**, 1-11 (2004).
16. Dai, A., Wigley, T. M. L. Global patterns of ENSO-induced precipitation. *Geophys Res Lett.* **27**, 1283–1286 (2000).
17. Sun X, Renard B, Thyer M, Westra S, Lang M. A global analysis of the asymmetric effect of ENSO on extreme precipitation. *J. Hydrol.* **530**, 51–65 (2015).
18. Sun, Q., Miao, C., AghaKouchak, A., Duan, Q. Century-scale causal relationships between global dry/wet conditions and the state of the Pacific and Atlantic Oceans. *Geophys Res Lett.* **43**, 6528–6537 (2016).
19. Rajeevan M, Bhate J, Jaswal AK (2008) Analysis of variability and trends of extreme rainfall events over India using 104 years of gridded daily rainfall data. *Geophys Res Lett.*
20. Yi, X. S., Li, G. S., Yin, Y. Y. Spatio-temporal variation of precipitation in the three-river headwater region from 1961 to 2010. *J. Geogr. Sci.* **23**, 447-464 (2013).
21. Chai, Y., Martins, G., Nobre, C. et al. Constraining Amazonian land surface temperature sensitivity to precipitation and the probability of forest dieback. *npj Clim Atmos Sci.* **4**, 6 (2021).
22. Hall, A., Cox, P., Huntingford, C., & Klein, S. (2019). Progressing emergent constraints on future climate change. *Nature Climate Change*, **9(4)**, 269-278.
23. Chen, J., Brissette, F. P., Leconte, R. Uncertainty of downscaling method in quantifying the impact of climate change on hydrology. *J. Hydrol.* **401**, 190-202 (2011).
24. Johnson, F., & Sharma, A. Accounting for interannual variability: a comparison of options for water resources climate change impact assessments. *Water Resour. Res.* **47**, W04508 (2011).
25. Hempel, S., Frieler, K., Warszawski, L., Schewe, J., & Piontek, F. A trend-preserving bias correction—the ISI-MIP approach. *Earth Syst. Dynam.* **4**, 219-236 (2013).

Responses to Reviewer #2

Review for the paper entitled "Constrained CMIP6 projections indicate slower warming rates and reduced water availability across Asia"

General Comments: The paper is well written and clear on the results. The methodology is explained in detail with all information needed. The findings are significant and seem supported by the figures, but I have a main major concern about their interpretation.

Response: Thank you very much. We have carefully revised the manuscript according to the referee's comments, which were very valuable for improving the interpretation of the findings.

Comment 1. Most of the emergent constraint signal is claimed to be related to precipitation feedback on temperatures. However, by reading the picture my first conclusion would be that models warming the most also see the largest increase in precipitation simply because of thermodynamics laws (higher temperature = more evaporative world). For me it's not really clear that there is a feedback here. Maybe one thing to do would be to express the precipitation change per °C (instead of per year).

That's my only major concern about the paper but I think it really needs to be clarified (either by clearly showing how it's a feedback and not just thermodynamics, or by rephrasing the text and discussion around thermodynamics considerations). It would probably not change the main results (the relation between model warming the most and having more precipitation would still be the same), but it must be clear why it is so.

Response: We agree with the referee that our findings are mostly due to the thermodynamics. We have clarified this in the revised version and rephrased the text throughout (Lines 119, 172, 313 in the revised main text). We have also expressed the precipitation change per °C in addition to the change per year (Please see Lines 9-11 and 167-171 in the revised main text).

Some other suggestions:

Comment 1. Line 65: "Asia" domain should be defined here maybe?

Response: Thank you for your comment. We have shown the coverage of Asia in this study in Fig. 1a in the revised main text. Supplementary Figures 2 and 7 in the updated SI are also replaced with figures showing the Asian domain.

Comment 2. L70: "Asian precipitation": do you consider yearly precip or only during the summer monsoon?

Response: It is yearly precipitation. We have replaced by "Asian annual precipitation". Please see Lines 69-70 in the revised main text.

Comment 3. L137-143: Maybe a K-S test (or other) could be used to see how significant is the shift between pdfs?

Response: We implemented the K-S test to check how significant the shift is between pdfs with reference to Cox et al. (2018)¹. We generated the cumulative distributions of the future precipitation growth rate before and after application of the emergent constraint (Supplementary Figure 11 in the revised SI), calculated the K-S statistics (See Supplementary Table 9 in the revised SI), and found the shifts are significant under a confidence level of 95% (all the K-S statistics are higher than the critical value at the 95% confidence level). We added relevant text in Lines 166-167 in the revised main text.

Comment 4. L164-170: It's not surprising to see a relationship between longwave and temperature, because longwave is basically temperature. Maybe you could look at the latent heat fluxes here too, which is more related to evaporation/precipitation.

Response: Following your suggestion, we have further investigated the relationship between latent heat fluxes and precipitation based on 25 CMIP6 model outputs (Supplementary Figure 18 in the revised SI). In addition, we also explored the potential emergent constraint relationships between historical trends in relative humidity/soil water content/land surface runoff and future annual precipitation growth rates across the CMIP6 models (see Lines 185-187 in the revised main text and Supplementary Figures 19 to 21 in the revised SI).

Comment 5. L191-195: Instead of constraining the future temperatures based on a constrained precipitation why not simply constrain future T based on historical T?

Response: Thank you for your suggestion. We set up the emergent constraint relationship between the annual growth rates of the historical temperature and the future temperature across 23 CMIP6 models (Supplementary Figure 23a), and explained the physical mechanisms as follows:

There exists a proportionally positive feedback in temperature to the rising radiative forcing, i.e., past and future warming trends are both controlled by the sensitivity of such feedback². Thus, the future warming trends are firmly linked with the historical temperature growth rate which can be used to constrain the future temperature growth rate. This mechanism has been widely applied to constrain equilibrium climate sensitivity (ECS), transient climate response (TCR) and ocean heat uptake^{2,3-7} (see Lines 215-221 in the revised main text).

The constrained results by using past warming trends are highly consistent with our original conclusions (see Supplementary Figure 23b and c, and Lines 221-227 in the revised main text).

Comment 6. Fig.1c: It would be nice to add some robustness/uncertainty on OBS trends (by bootstrapping the sampling for example and recomputing the trend...)

Response: Yes, and we have added robustness/uncertainty on OBS trends by bootstrapping the sampling and recomputing the trends (see the revised Figure captions of Fig. 1d in Lines 506-508).

References

1. Cox, P., Huntingford, C. & Williamson, M. Emergent constraint on equilibrium climate sensitivity from global temperature variability. *Nature*. **553**, 319-322 (2018).
2. Tokarska, K. B., Stolpe, M. B., Sippel, S., Fischer, E. M., & Knutti, R. Past warming trend constrains future warming in CMIP6 models. *Sci. Adv.* **6**, eaaz9549 (2020).
3. Knutti, R., Rugenstein, M. & Hegerl, G. Beyond equilibrium climate sensitivity. *Nature. Geosci.* **10**, 727–736 (2017).
4. Jiménez-de-la-Cuesta, D., Mauritsen, T. Emergent constraints on Earth’s transient and equilibrium response to doubled CO₂ from post-1970s global warming. *Nat. Geosci.* **12**, 902–905 (2019).
5. Armour, K. C. Energy budget constraints on climate sensitivity in light of inconstant climate feedbacks. *Nat. Clim. Chang.* **7**, 331–335 (2017).
6. Monckton, C., Soon, W. W.-H., Legates, D. R. & Briggs, W. M. Why models run hot: results from an irreducibly simple climate model. *Sci. Bull.* **60**, 122–135 (2015).
7. Otto, A. et al. Energy budget constraints on climate response. *Nat. Geosci.* **6**, 415–416 (2013).

Responses to Reviewer #3

General Comments: With a similar approach done by Chai et al. (2021) for the Amazon dieback, this manuscript investigated historical temperature and precipitation trends in Asia by CMIP6 models against the observations, and used the results to adjust future projections. As CMIP6 models overestimate historical precipitation changes during 1970–2014, future precipitation projections are downgraded. However, the reviewer has some concerns.

Response: Thank you for your helpful comments which were very valuable for improving the manuscript. We have carefully revised the manuscript to address each of the referee's concerns.

Comment 1. First, the emergent constraint seeks physical mechanism of a possible relationship between the two variables, but this approach will not work well in this subject, because regional precipitation is not governed by temperatures only, but also much influenced by circulation changes and associated moisture convergence.

Response: We agree with the referee that it is important to assess the degree to which changes in regional precipitation are governed by temperature changes vs circulation changes and associated moisture convergence. To examine whether the influence of climate variability (such as the monsoon, ENSO, AO, etc.) alters the thermodynamic precipitation-temperature relationship in different regions of Asia on annual timescales, first, we randomly selected eight square areas of Asia located at different latitudes (see Supplementary Figure 5 in the revised SI and Lines 91-94 in the revised main text), and found that the significant positive relationships during 1970–2014 were consistent in each of the sub-areas based on various observed datasets of precipitation (GPCC, 20CRv2c, HadCRUT4, GHCN, CMAP and ERA-Interim) and temperature (Delaware, HadCRUT4, GISS, NOAA). Furthermore, we examined the spatial distribution of the correlation coefficient during the future period (2015–2100) based on the CMIP6 projections, and found high positive correlation coefficients ($R \geq 0.4$ and corresponding P value < 0.001) between temperature and precipitation across 82.0–92.1% of Asia (see Supplementary Figure 7 in the revised SI and Lines 96-97 in the revised main text). Therefore, the linear relationship is overall robust over Asia, and the thermodynamic effect is equally important across the whole of Asia.

Second, considering that circulation changes, such as monsoons, ENSO and AO, etc. usually affect the relationship between temperature and precipitation by increasing the likelihood and intensity of extreme weather events¹⁻⁴, we used the moving average method with window lengths of 5–10 years to examine the precipitation-temperature relations. With this method, the influence of large-scale climate variability is significantly reduced and the long-term trend is better reflected⁵⁻⁶. The results show that strong positive relations still exist between precipitation and surface air temperature after smoothing out extreme fluctuations (see Supplementary Figure 8a in the revised SI), proving the reliability of the previously identified relationships. Furthermore, the sensitivity of precipitation to temperature change estimated by the new relationship remains (it is only slightly increased by 8.9–10.1%, see Supplementary Figure 8b in the revised SI), implying that the effects of monsoons, ENSO and AO are not significant in this study. We added relative explanations in the revised manuscript (see Lines 126-137 in the revised main text).

Finally, we provided further evidence to support the linear relationships by collecting additional observed datasets of precipitation (GPCC, 20CRv2c,

HadCRUT4, GHCN, CMAP and ERA-Interim) and temperature (Delaware, HadCRUT4, GISS, NOAA) in Asia during 1970–2014, and by extracting future projections (2015–2100) of precipitation and temperature in Asia for each of the 27 CMIP6 models. Consistently, significant linear relationships have been identified among all the datasets (see Supplementary Figure 4 in the revised SI and Fig. 1e in the main text and Lines 87-91 in the revised main text) and in almost all the models under the four SSPs (see Supplementary Figure 6 in the revised SI and Lines 94-96 in the revised main text). The only exception is the NorESM2-LM model which exhibits poor performance in reproducing historical precipitation under SSP126 (the correlation coefficient between simulated and observed precipitation during 1970-2015 is only of 0.09, and the P value is 0.56). In addition, in-depth explanations for the physical mechanisms behind the constraint relationship have been added (see Lines 110-121 in the revised main text and Lines 357-374 in Method).

Comment 2. Second, the authors adjust future precipitation changes in the models in which historical temperature growth rate is large, although the ensemble mean historical temperature trend fits the observation very well.

Response: We added a new figure (see Supplementary Figure 10 in the revised SI) to demonstrate that even if the ensemble mean historical temperature trend fits the observations very well, the constrained future precipitation may still differ from the raw projections with different levels of uncertainty. This is mainly due to different probability distributions of observational temperature and simulated temperature. We clarified this in Lines 150-153 in the revised main text.

Comment 3. Third, after reducing the future precipitation trends, authors decrease the future temperature trends, although models had no significant bias in their historical period. The reviewer is not confident with the authors hypotheses.

Response: Indeed, the mean temperature of the models shows no significant bias against observations in the historical period. However, individual models may still perform poorly (see Supplementary Figure 22 in the revised SI and Lines 196-199 in the revised main text), leading to large uncertainties in simulating the temperature growth rate in both the historical and the future periods (see Fig. 1b). Therefore, it is legitimate to wish to constrain the model projections.

To further verify the constrained results of the future temperature growth rate, we set up an emergent constraint relationship between the annual growth rates of the historical temperature and the future temperature across 23 CMIP6 models (Supplementary Figure 23a), considering the firm link between the past and future warming trends (Tokarska et al., 2020⁷). The physical mechanism behind the potential constrained relationships is as follows:

There exists a proportionally positive feedback in temperature to the rising radiative forcing, i.e., the past and future warming trends are both controlled by the sensitivity of such feedback². Thus, the future warming trends are firmly linked with the historical temperature growth rate, which can be used to constrain the future temperature growth rate. This mechanism has been widely applied to constrain equilibrium climate sensitivity (ECS), transient climate response (TCR) and ocean heat uptake⁸⁻¹¹ (see Lines 215-221 in the revised main text).

The new constrained results are highly consistent with our original conclusions (see Supplementary Figure 23b and c, and Lines 221-227 in the revised main text).

Other comments

Comment 1. Methods for calculation are missing. How are different horizontal resolutions of observations and models treated? Which is the area of Asia?

Response: Thank you for your comments. We re-gridded all the CMIP6 output layers and the observational data sets to a uniform $0.25^{\circ} \times 0.25^{\circ}$ latitude-longitude spatial resolution for calculating the multi-model mean values (see Lines 348-350 in the revised Data Availability section). Additionally, we have added the coverage of Asia applied in this study in Fig. 1a in the revised main text. Supplementary Figures 2 and 7 in the updated SI are also replaced with the Asian domain versions.

Comment 2. Line 155-159: Over the Amazon, is physiological response the main reason of precipitation decrease in future, rather than an El Nino-like mean SST changes?

Response: As the referee pointed out, the reasons for the precipitation decrease over the Amazon in the future are very complicated. Some studies held that physiological responses are the main reasons for the future decreased precipitation (Kooperman et al., 2018¹²; Richardson et al., 2018¹³; Langenbrunner et al., 2019¹⁴). However, other studies reported that El Nino-like mean SST changes will also bring large effects (Martins et al., 2015)¹⁵. In the revised manuscript, we have added discussions on the effects of El Nino-like mean SST changes on precipitation in Amazon (see Lines 181-182 in the revised main text).

Comment 3. What is the difference between Supplementary Table 1 and 2? Captions and contents are same.

Response: Thanks for pointing this out. We have removed the Supplementary Table 2 in the revised SI.

References

1. Dai, A, Wigley, T. M. L. Global patterns of ENSO-induced precipitation. *Geophys Res Lett.* **27**, 1283-1286 (2000).
2. Sun X, Renard B, Thyer M, Westra S, Lang M. A global analysis of the asymmetric effect of ENSO on extreme precipitation. *J. Hydrol.* **530**, 51-65 (2015).
3. Sun, Q., Miao, C., AghaKouchak, A., Duan, Q. Century-scale causal relationships between global dry/wet conditions and the state of the Pacific and Atlantic Oceans. *Geophys Res Lett.* **43**, 6528-6537 (2016).
4. Rajeevan M, Bhate J, Jaswal AK (2008) Analysis of variability and trends of extreme rainfall events over India using 104 years of gridded daily rainfall data. *Geophys Res Lett.*
5. Yi, X. S., Li, G. S., Yin, Y. Y. Spatio-temporal variation of precipitation in the three-river headwater region from 1961 to 2010. *J. Geogr. Sci.* **23**, 447-464 (2013).
6. Chai, Y., Martins, G., Nobre, C. et al. Constraining Amazonian land surface temperature sensitivity to precipitation and the probability of forest dieback. *npj Clim Atmos Sci.* **4**, 6 (2021).

7. Tokarska, K. B., Stolpe, M. B., Sippel, S., Fischer, E. M., & Knutti, R. Past warming trend constrains future warming in CMIP6 models. *Sci. Adv.* **6**, eaaz9549 (2020).
8. Jiménez-de-la-Cuesta, D., Mauritsen, T. Emergent constraints on Earth's transient and equilibrium response to doubled CO₂ from post-1970s global warming. *Nat. Geosci.* **12**, 902–905 (2019).
9. Armour, K. C. Energy budget constraints on climate sensitivity in light of inconstant climate feedbacks. *Nat. Clim. Chang.* **7**, 331–335 (2017).
10. Monckton, C., Soon, W. W.-H., Legates, D. R. & Briggs, W. M. Why models run hot: results from an irreducibly simple climate model. *Sci. Bull.* **60**, 122–135 (2015).
11. Otto, A. et al. Energy budget constraints on climate response. *Nat. Geosci.* **6**, 415–416 (2013).
12. Kooperman, G.J., Chen, Y., Hoffman, F.M. et al. Forest response to rising CO₂ drives zonally asymmetric rainfall change over tropical land. *Nature. Clim. Change.* **8**, 434–440 (2018).
13. Richardson, T., Forster, P., Andrews, T., Boucher, O., Faluvegi, G., Fläschner, D., Kasoar, M., Kirkevåg, A., Lamarque, J. F., Myhre, G., Olivié, D., Samset, B. H., Shawki, D., Shindell, D., Takemura, T., & Voulgarakis, A. Carbon dioxide physiological forcing dominates projected eastern Amazonian drying. *Geophys. Res. Lett.* **45**, 2815–2825 (2018).
14. Langenbrunner, B., Pritchard, M. S., Kooperman, G. J., & Randerson, J. T. Why does Amazon precipitation decrease when tropical forests respond to increasing CO₂?. *Earths. Future.* **7**, 450-468 (2019).
15. Martins, G., Randow, C. V., Sampaio, G., & Dolman, A. J. Precipitation in the Amazon and its relationship with moisture transport and tropical pacific and Atlantic SST from the CMIP5 simulation. *Hydrol. Earth. Syst. Sci. Discuss.* **12**, 671-704 (2015).

REVIEWER COMMENTS

Reviewer #1 (Remarks to the Author):

The authors properly revised the manuscript according to my comments. Therefore, it is acceptable for the publication in Nature Communications.

Reviewer #2 (Remarks to the Author):

Authors have respond clearly to all my comments and updated the manuscript accordingly. Thus I find it suitable for publication.

Reviewer #3 (Remarks to the Author):

I still have the same comments raised to the original manuscript. In particular, I do not agree applying an emergent constraint between temperature and precipitation to future regional precipitation changes in Asia because many factors other than thermal conditions would work on regional precipitation changes. Also, future temperature constraint is not understandable because "the mean temperature of the models shows no significant bias against observations in the historical period". The reviewer considers that a rebuttal on these points has not been properly made.

Responses to Reviewers #1 and #2

Reviewer #1 (Remarks to the Author):

The authors properly revised the manuscript according to my comments. Therefore, it is acceptable for the publication in Nature Communications.

Reviewer #2 (Remarks to the Author):

Authors have responded clearly to all my comments and updated the manuscript accordingly. Thus, I find it suitable for publication.

Response: Thank you very much for your constructive comments and suggestions in the previous round. In this round, we believe we have fully addressed your further concerns about the physical mechanisms behind the emergent constraint between temperature and precipitation in Asia. Please refer to our detailed responses to the Editor and Reviewer #3.

Responses to Reviewer #3

Comment 1: I still have the same comments raised to the original manuscript. In particular, I do not agree applying an emergent constraint between temperature and precipitation to future regional precipitation changes in Asia because many factors other than thermal conditions would work on regional precipitation changes.

Response: As the referee rightly points out, it is fundamentally important to confirm that the emergent relationship between the growth rates of temperature and precipitation arises from physical processes, rather than emerging by chance, especially at the continental scale of Asia. To confirm that an emergent constraint actually exists, Hall et al. (2019)¹ and Williamson et al. (2021)² have proposed two indicators: (1) plausible mechanisms (or verification of mechanisms); and (2) out-of-sample testing.

(1) Plausible mechanisms.

The referee points out that many factors other than thermal conditions would affect regional precipitation – which is entirely reasonable. Previous studies mainly classified factors affecting precipitation into ‘dynamic’ and ‘thermodynamic’ categories (Emori and Brown, 2005³; Pfahl et al., 2017⁴; Ali and Mishra, 2018⁵). Two variables have been widely used to measure the dynamic conditions, i.e., pressure vertical velocity as an indicator of the strength of dynamic disturbance (Emori and Brown, 2005³; Pfahl et al., 2017⁴), and convective available potential energy (CAPE) as an indicator of atmospheric instability and convection strength (Seeley and Romps, 2015⁸). These two variables are commonly used to explore atmospheric dynamic impacts on precipitation extremes. Inspired by the reviewer’s insightful comments, we also attempt to explore whether long-term changes in mean precipitation are governed by these dynamic drivers. First, we test the relationship between annual precipitation and the annual mean pressure vertical velocity at 11 pressure levels in Asia based on the ERA5 reanalysis data during 1979-2014, and find poor correlation (Figure R3, $-0.46 < r < -0.26$, $p > 0.01$). Then, we also test the CAPE relationship with annual precipitation in Asia during 1979-2014, and again find poor correlation (Figure R4, $r = -0.24$, $p > 0.1$). These results imply that ‘dynamic’ factors do not strongly affect long-term mean precipitation over the large area considered in our study.

Figure R3. Poor correlation between pressure vertical velocity and precipitation at 11 pressure levels and precipitation in Asia during 1979-2014.

Figure R4. Poor correlation between convective available potential energy and precipitation in Asia during 1979-2014 based on ERA5 dataset.

Emori and Brown (2005)³ suggested that dynamic change provides only a partial explanation for the increase in annual mean precipitation in the tropical Pacific (belonging to its oceanic area), whereas thermodynamic change explains almost all the increase at mid- to high-latitudes (including Asia) based on model experiments for 2081-2100. A stochastic simulation of large-scale condensation, carried out by O’Gorman and Schneider (2008)⁹, based on a partial differential equation that includes water vapor transport and thermodynamics processes (Equation 11 in O’Gorman and Schneider, 2008⁹), showed that large-scale (i.e. the grid scale used by O’Gorman and Schneider, 2008⁹) precipitation increases approximately linearly with temperature over a wide range of temperatures (Figure 7 in O’Gorman and Schneider, 2008⁹). We acknowledge that the ‘dynamic’ factors play an important role in altering precipitation extremes at regional scale (Pfahl et al., 2017⁴). Nonetheless, our study focuses on long-term changes in mean precipitation; previous studies have highlighted divergent responses of mean and extreme precipitation to climate change (Emori and Brown, 2005³). At the time of writing, it is very difficult to find evidence to support the proposition that dynamic factors significantly drive such long-term changes in mean precipitation over continental Asia. Given that the ‘thermodynamic’ factors have been widely recognized as playing the lead role in driving changes in long-term mean precipitation over large areas (Emori and Brown, 2005³; O’Gorman and Schneider, 2008⁹), we leave the dynamic attributions to future work (we state this in the summary paragraph in the main text, see Lines 271-275).

Taking the reviewer’s comments fully into consideration, we also analyze whether the relationship between the growth rates of simulated historical temperature and projected future precipitation in Asia is significantly affected by regions experiencing strong atmospheric circulation. According to Van der Ent et al. (2010)¹⁰, precipitation in regions with low continental precipitation-recycling ratio (ρ_c , defined as the ratio of precipitation of continental origin [not necessarily from the same continent] to that of both continental and oceanic origin) primarily derives from moisture transport from the oceans. Here we selected areas with $\rho_c < 0.5$ in Asia as regions subject to monsoons and ENSO events (such as southeastern China, etc.). By examining the relationship between the growth rates of simulated historical temperature and projected future precipitation when excluding regions of low ρ_c , we found that the linear regression results remain almost unchanged under all four shared socioeconomic pathways (SSPs) (i.e., changes in the regression slope are only 0.2–16.0%, see Figure R5). This implies that such

regions with strong atmospheric circulation in Asia do not affect the overall relationship between the long-term mean growth rates of precipitation and temperature in Asia.

Figure R5. Relationships between historical temperature growth rates and future precipitation growth rates in Asia, with (red fitted lines) and without (blue fitted lines) regions of low precipitation-recycling ratio ($\rho_c < 0.5$), for the following emission scenarios: (a) SSP126; (b) SSP245; (c) SSP370; and (d) SSP585.

A similar analysis using a moving average method with window lengths of 5–10 years to exclude extreme precipitation mainly driven by monsoons and ENSO events was conducted in the previous round of revision. The findings were similar in that strong positive relations persist (with slight increases by 8.9–10.1%, see Figure R6) between precipitation and surface air temperature after smoothing out extreme fluctuations. These again support the physical hypothesis that precipitation change in Asia is not so affected by monsoons and ENSO events.

We added the above explanations on mechanisms in Lines 122-129 and Lines 271-275 in the main text, Section 2 in the Supplementary Text, and Supplementary Figs. 8-11.

Figure R6. Linear relationships between precipitation and land surface air temperature during 1970–2014. (a) presents linear precipitation–temperature relationships obtained for the initial observations (Raw_observation) and for observations smoothed by a series of moving windows (with window lengths ranging from 5 to 10 years). Each circle represents a year. (b) indicates the sensitivity of precipitation to temperature (slope values in a) for the initial observations and for the observations smoothed using a series of moving windows.

(2) Out-of-sample testing.

Hall et al. (2019)¹ pointed out that testing the emergent constraint relationship using other ESM ensembles is an indirect but effective way to improve the reliability of introduced emergent constraints because it is equivalent to enlarging the original ensemble. Provided high correlation persists, the likelihood that the emergent relationship has emerged by chance is then greatly reduced. We therefore also tested the emergent relationships in the CMIP5 models. As shown in Figure R7, the relationship between temperature and precipitation also exists across CMIP5 models achieving high correlation ($r=0.72$, p value <0.001). We added relevant contents in Lines 139-142 in the main text, and Supplementary Fig. 13.

Figure R7. Relationship between growth rates of historical temperature and future precipitation in Asia based on CMIP5 projections under the RCP4.5 emission scenario. Each circle represents a CMIP5 model.

In summary, we have confirmed the emergent constraint relationship in our study by fully analyzing the two confirmation indicators proposed by Hall et al. (2019)¹. However, as Hall et al. (2019)¹ pointed out: “in practice it is likely no EC (i.e. emergent constraint) can ever be completely confirmed and is instead associated with degrees of confirmation.” We believe that the degree of confirmation of our emergent constraint relationship is already very high, compared to that of other emergent constraint papers.

Comment 2: Also, future temperature constraint is not understandable because “the mean temperature of the models shows no significant bias against observations in the historical period”. The reviewer considers that a rebuttal on these points has not been properly made.

Response: Thank you for pointing this out. The reviewer is correct that the multi-model mean value of annual average land surface air temperature in Asia (4.43 °C) shows no significant bias against observations (4.44 °C) during 1970–2014. However, in our case, we try to bring a constraint on annual growth rate in temperature, rather than the mean temperature. In fact, the multi-model mean annual growth rate in temperature (0.363 ± 0.0732 °C decade⁻¹) has non-negligible bias in comparison with observations (0.326 ± 0.035 °C decade⁻¹); the simulated temperature growth rate is therefore overestimated by 11.35%. Probability density distributions of annual growth rates in temperature based on observations (red line in Figure R8) and the CMIP6 multi-model mean values

(black line in Figure R8) also exhibit large discrepancy. Such obvious bias indicates the necessity to bring a constraint on future temperature growth rate. The constrained results of future temperature growth rate also exhibit an overestimate of 3.4–11.6% compared with the raw projection.

To avoid confusion, we have included a discussion of the discrepancy between the observed and simulated temperature growth rates in the revised manuscript (see Lines 193-199 in the main text).

Figure R8. Probability density distributions of annual growth rate in temperature based on observations (red line, $0.0326 \pm 0.0035 \text{ } ^\circ\text{C year}^{-1}$) and CMIP6 multi-model mean values (black line, $0.0363 \pm 0.00732 \text{ } ^\circ\text{C year}^{-1}$).

References

1. Hall, A., Cox, P., Huntingford, C., & Klein, S. (2019). Progressing emergent constraints on future climate change. *Nature Climate Change*, 9(4), 269-278.
2. Williamson, M. S., Thackeray, C. W., Cox, P. M., Hall, A., Huntingford, C., & Nijse, F. J. (2021). Emergent constraints on climate sensitivities. *Reviews of Modern Physics*, 93(2), 025004.
3. Emori, S. & Brown, S. J. (2005). Dynamic and thermodynamic changes in mean and extreme precipitation under changed climate. *Geophysical Research Letters*, 32(17), L17706.
4. Pfahl, S., O’Gorman, P. A. & Fischer, E. M. (2017). Understanding the regional pattern of projected future changes in extreme precipitation. *Nature Climate Change*, 7, 432-427.
5. Ali, H. & Mishra, V. (2018). Contributions of dynamic and thermodynamic scaling in subdaily precipitation extremes in India. *Geophysical Research Letters*, 45, 2352-2361.

6. Gorodetskaya, I. V., Tremblay, L. B., Liepert, B., Cane, M. A. & Cullather, R. I. (2008). The influence of cloud and surface properties on the arctic ocean shortwave radiation budget in coupled models. *Journal of Climate*, 21(5), 866-882.
7. Wenzel, S., Cox, P. M., Eyring, V., & Friedlingstein, P. (2016). Projected land photosynthesis constrained by changes in the seasonal cycle of atmospheric CO₂. *Nature*, 538(7626), 499-501.
8. Seeley, J. T. & Romps, D. M. (2015). Why does tropical convective available potential energy (cape) increase with warming? *Geophysical Research Letters*, 42, 10429-10437.
9. O'Gorman, P. A., & Schneider, T. (2008). The hydrological cycle over a wide range of climates simulated with an idealized GCM. *Journal of Climate*, 21(15), 3815-3832.
10. Van der Ent, R. J., Savenije, H. H., Schaefli, B., & Steele-Dunne, S. C. (2010). Origin and fate of atmospheric moisture over continents. *Water Resources Research*, 46(9).

REVIEWERS' COMMENTS

Reviewer #2 (Remarks to the Author):

I think the authors have made a significant effort to answer all comments from Reviewer 3. It seems for me that their analysis is convincing and justified enough to be published.

Reviewer #3 (Remarks to the Author):

I appreciated further analysis made by authors on relative importance between thermodynamic and dynamic contribution on future mean precipitation changes. My impression from Figure R3 is that dynamic factor with $r \sim 0.5$ may not be neglected. They may be significant at 95% level. I wish the authors make the dynamic attributions in their future work. As this is reminded explicitly in the revised main text and further evaluation of this paper will take place once it is made public, I agree on a publication of this manuscript.

Responses to Reviewers' comments

Reviewer #2 (Remarks to the Author):

Comment: I think the authors have made a significant effort to answer all comments from Reviewer 3. It seems for me that their analysis is convincing and justified enough to be published.

Response: Thank you very much for reviewing our manuscript. Your comments during the whole process helped to improve our paper considerably.

Reviewer #3 (Remarks to the Author):

Comment 1: I appreciated further analysis made by authors on relative importance between thermodynamic and dynamic contribution on future mean precipitation changes.

Response: The authors appreciate the reviewer's comments during the whole review process, especially the one concerning the impact of dynamic factors which could be negligible at regional scale. By addressing your previous concerns, our study has become more convincing.

Comment 2: My impression from Figure R3 is that dynamic factor with $r \sim 0.5$ may not be neglected. They may be significant at 95% level. I wish the authors make the dynamic attributions in their future work. As this is reminded explicitly in the revised main text and further evaluation of this paper will take place once it is made public, I agree on a publication of this manuscript.

Response: The reviewer is right that dynamic factors with $r \sim 0.5$ may be significant at the 95% level and must be assessed in future work. Thus, for clarity, in the revised manuscript, we indicate on the figure panels (see Supplementary Fig. 8) whether the correlations between pressure vertical velocity at 11 pressure levels and precipitation are significant at the 95% level ($0.01 < p \text{ value} < 0.05$), or not ($p \text{ value} > 0.05$). We changed Lines 141-143 and 146-147 to “we found that the dynamic factors

exhibit some correlation with the long-term trend in precipitation in continental Asia ($-0.46 < r < -0.24$), but not so strong as the thermodynamic factors ... However, the contribution of dynamic factors will be assessed in future work”, and Lines 290-295 to “Although thermodynamic factors have been widely recognized as playing the lead role in driving changes in long-term mean precipitation over large areas⁶⁵⁻⁶⁶ (while vertical pressure velocity and CAPE have smaller correlation coefficients), dynamic factors may still be significant (Supplementary Fig. 8) under certain circumstances. Therefore, it would be worthwhile to determine the specific contributions of the dynamic factors to the long-term trend in precipitation change at the continental scale in future work.”